# FANCM promotes class I interfering crossovers and suppresses class II non-interfering crossovers in wheat meiosis

Stuart D. Desjardins[1], James Simmonds [2], Inna Guterman[1], Kostya Kanyuka [3,4], Amanda J. Burridge [5], Andrew J. Tock [6], Eugenio Sanchez-Moran[7], F. Chris H. Franklin[7], Ian R. Henderson [6], Keith J. Edwards[5], Cristobal Uauy [2] & James D. Higgins [1✉]

FANCM suppresses crossovers in plants by unwinding recombination intermediates. In wheat, crossovers are skewed toward the chromosome ends, thus limiting generation of novel allelic combinations. Here, we observe that FANCM maintains the obligate crossover in tetraploid and hexaploid wheat, thus ensuring that every chromosome pair exhibits at least one crossover, by localizing class I crossover protein HEI10 at pachytene. FANCM also suppresses class II crossovers that increased 2.6-fold in *fancm msh5* quadruple mutants. These data are consistent with a role for FANCM in second-end capture of class I designated crossover sites, whilst FANCM is also required to promote formation of non-crossovers. In hexaploid wheat, genetic mapping reveals that crossovers increase by 31% in *fancm* compared to wild type, indicating that *fancm* could be an effective tool to accelerate breeding. Crossover rate differences in *fancm* correlate with wild type crossover distributions, suggesting that chromatin may influence the recombination landscape in similar ways in both wild type and *fancm*.

[1] Department of Genetics and Genome Biology, Adrian Building, University of Leicester, University Road, Leicester LE1 7RH, UK. [2] Department of Crop Genetics, John Innes Centre, Norwich NR4 7UH, UK. [3] NIAB, 93 Lawrence Weaver Road, Cambridge CB3 0LE, UK. [4] Biointeractions and Crop Protection, Rothamsted Research, Harpenden AL5 2JQ, UK. [5] Life Sciences Building, University of Bristol, 24 Tyndall Avenue, Bristol BS8 1TQ, UK. [6] Department of Plant Sciences, University of Cambridge, Downing Street, Cambridge CB2 3EA, UK. [7] School of Biosciences, University of Birmingham, Birmingham B15 2TT, UK. ✉email: jh555@leicester.ac.uk

Meiotic recombination is initiated by numerous programmed DNA double-strand breaks (DSBs), catalyzed by SPO11 and MTOPVIB[1,2], that are repaired by homologous recombination into crossovers (COs) or non-crossovers (NCOs). COs are characterized by the reciprocal exchange of DNA between homologous chromosomes, whereas NCOs occur at sites of non-reciprocal DNA repair using either the homologue or the sister-chromatid as a template[3]. In Arabidopsis and wheat, COs form via the class I or class II pathways[4–7]. The class I pathway accounts for ~85% COs and is required to maintain CO assurance, thus formation of the obligate CO. The class I pathway ensures that each chromosome pair receives at least one CO through the action of meiosis specific proteins MER3/RCK, MSH4, MSH5, SHOC1, ZIP4, PTD and HEI10[6], thereby promoting correct chromosome segregation. Class I COs are interference-sensitive and therefore more likely to be spaced further apart than would be expected by random chance[8–10]. Class II COs are interference-insensitive, accounting for ~15% COs and are partially dependent on the MUS81 endonuclease[11,12].

The number of COs per chromosome is tightly constrained in the majority of eukaryotes (~1–3), regardless of physical chromosome size[5,13]. DSBs usually occur far in excess of COs in plants, suggesting that they do not limit CO formation. In hexaploid wheat the skewed ratio of ~2100 DSBs to ~42 COs per cell reveals that only 2% of potential repair sites mature into COs[14], indicating that underlying mechanisms negatively regulate CO formation. In Arabidopsis, class I COs are limited by expression of HEI10[15], the presence of PPX1[16], and the installation of the synaptonemal complex transverse filament proteins ZYP1a/ZYP1b[17,18]. Class II COs are limited by three independent, anti-recombination pathways involving FANCM, FIDGETIN and RECQ4[19–22]. The FANCM (Fanconi anemia complementation group M homolog) helicase functions with MHF1 and MHF2 to limit COs by unwinding inter-homolog repair intermediates, such as D-loops, and promotes repair as NCOs, in inbred lines[19–21,23,24]. Ablation of FANCM in Arabidopsis restores bivalent formation in class I CO deficient mutants to near wild-type levels[19,20]. Single fancm mutants exhibited a 3-fold increase in COs by pollen fluorescent marker analysis[19], but chiasmata (the cytological manifestations of COs) were significantly reduced[20], although technical limitations may have precluded detection of more than one closely spaced CO using this analysis. A significant reduction in the number of bivalents and loss of the obligate chiasma was recently observed in lettuce fancm mutants[25], although Brassica (diploid and allotetraploid), rice (Oryza sativa) and pea (Pisum sativum) fancm mutants displayed increased COs[26,27].

Wheat is an allopolyploid crop species where the predominantly distally distributed COs/chiasmata rarely exceed two per chromosome pair[7,14,28,29]. COs mainly form in gene-rich regions[30] that contain a high frequency of TIR-Mariner transposons[31] and lower levels of DNA polymorphism[32]. COs also correlate with enrichment of the Polycomb histone modification H3K27me3 at the distal ends, indicating a potential role for facultative heterochromatin in shaping the recombination landscape[33]. CO frequency and distribution reduce the probability of creating advantageous novel allelic combinations in crop breeding programmes. Therefore, anti-CO factors, such as FANCM provide a potential way to overcome these limitations[34]. Here, using wheat fancm null mutants and VIGS, we demonstrate that class I COs are reduced in number, resulting in loss of the obligate CO, but this is offset by an increase in class II COs.

## Results

**FANCM is conserved in wheat.** To identify FANCM orthologues in wheat, BLAST searches were performed using the Arabidopsis thaliana FANCM amino acid sequence. Two FANCM orthologues were identified in tetraploid wheat T. turgidum: TtFANCM-A1 (TRITD4Av1G171480) and TtFANCM-B1 (TRITD4Bv1G035000). Three FANCM orthologues were identified in hexaploid wheat T. aestivum: TaFANCM-A1 (TraesCS4A02G217700), TaFANCM-B1 (TraesCS4B02G096400) and TaFANCM-D1 (TraesCS4D02G092800). All copies are full-length and predicted to produce functional proteins. FANCM is highly conserved between ploidy levels with FANCM-A1 and FANCMB-1 primary amino acid sequences being identical in tetraploid and hexaploid wheat (1447/1447 & 1458/1458, respectively), while the three FANCM homoeologues, FANCM-A1, FANCMB-1 and FANCM-D1, share 94.2% amino acid identity, with polymorphisms at 85 residues (1377/1462). A consensus sequence was created from the three wheat homoeologues to compare with Arabidopsis. Wheat FANCM shares 38.8% (543/1502) overall amino acid identity with Arabidopsis, with increased homology in the predicted DEXDc (DEAD-like helicases superfamily; 70.7%; 135/191) and HELICc (helicase superfamily c-terminal; 67.9%; 93/137) domains.

**FANCM is required for normal fertility.** Single fancm homoeolog mutants for tetraploid (Triticum turgidum, AABB) (Fig. 1a) and hexaploid wheat (Triticum aestivum, AABBDD) (Fig. 1b) were crossed to create null knockouts. The tetraploid Ttfancm_1 null exhibited a 36% reduction in total number of seeds per plant from $66 \pm 2$ ($n = 4$) in wild type to $43 \pm 5$ ($n = 4$) in the mutant ($p < 0.05$, Mann–Whitney U Test; Supplementary Fig. 1). The hexaploid Tafancm null exhibited a 15% reduction in seeds per plant from $164 \pm 4$ ($n = 6$) in wild type to $139 \pm 6$ ($n = 6$) in the mutant ($p < 0.05$, Mann–Whitney U Test; Supplementary Fig. 1). Pollen viability decreased from 92% in wild type ($n = 2042$) to 76% in Ttfancm_1 ($n = 1994$) (Supplementary Fig. 2). This indicates that defects occurring during gamete formation had a direct impact on seed development.

**FANCM is required for crossover assurance.** To determine the cause for the reduction in fertility and pollen viability in fancm null mutants, a cytological analysis was performed on tetraploid wheat meiocytes. At meiotic metaphase I (MI) the number of chiasmata, rod and ring bivalents was indistinguishable between wild type and the single homoeolog mutants (Ttfancm-A1_m1, Ttfancm-A1_m2, and Ttfancm-B1; p values > 0.05, Pairwise Wilcoxon Rank Sum Tests). However, in Ttfancm_1 nulls, chromosome pairs without chiasmata at MI revealed loss of the obligate chiasma and defective CO assurance (Fig. 1c,d, Supplementary Fig. 3, Supplementary Data 3). In Ttfancm_1, $0.98 \pm 0.09$ ($n = 104$) pairs of univalents per cell were observed at MI, compared to only $0.05 \pm 0.03$ ($n = 60$) in wild type ($p < 0.001$, Pairwise Wilcoxon Rank Sum Test), and 64% of cells possessed at least one univalent pair in Ttfancm_1, compared with only 5% in wild type. The number of rod bivalents per cell also increased from $1.7 \pm 0.16$ ($n = 60$) in wild type to $4.7 \pm 0.09$ ($n = 104$) in Ttfancm_1 ($p < 0.001$, Pairwise Wilcoxon Rank Sum Test). The increase in univalent pairs and rod bivalents coincided with an 18% decrease in chiasmata per cell, from $26 \pm 0.2$ ($n = 60$) in wild type to $22 \pm 0.3$ ($n = 104$) in Ttfancm_1 ($p < 0.001$, Pairwise Wilcoxon Rank Sum Test). There was no significant difference between Ttfancm_1 and Ttfancm_2 null mutants at MI (p values > 0.05, Pairwise Wilcoxon Rank Sum Tests). Chromosome pairs without chiasmata mis-segregated in Ttfancm_1, resulting in unbalanced gametes (Supplementary Fig. 4). At anaphase I, chromosome mis-segregation was observed in 43% of cells in Ttfancm_1 ($n = 40$), compared with 2.5% in wild type ($n = 40$). Chromosome mis-segregation through loss of the obligate

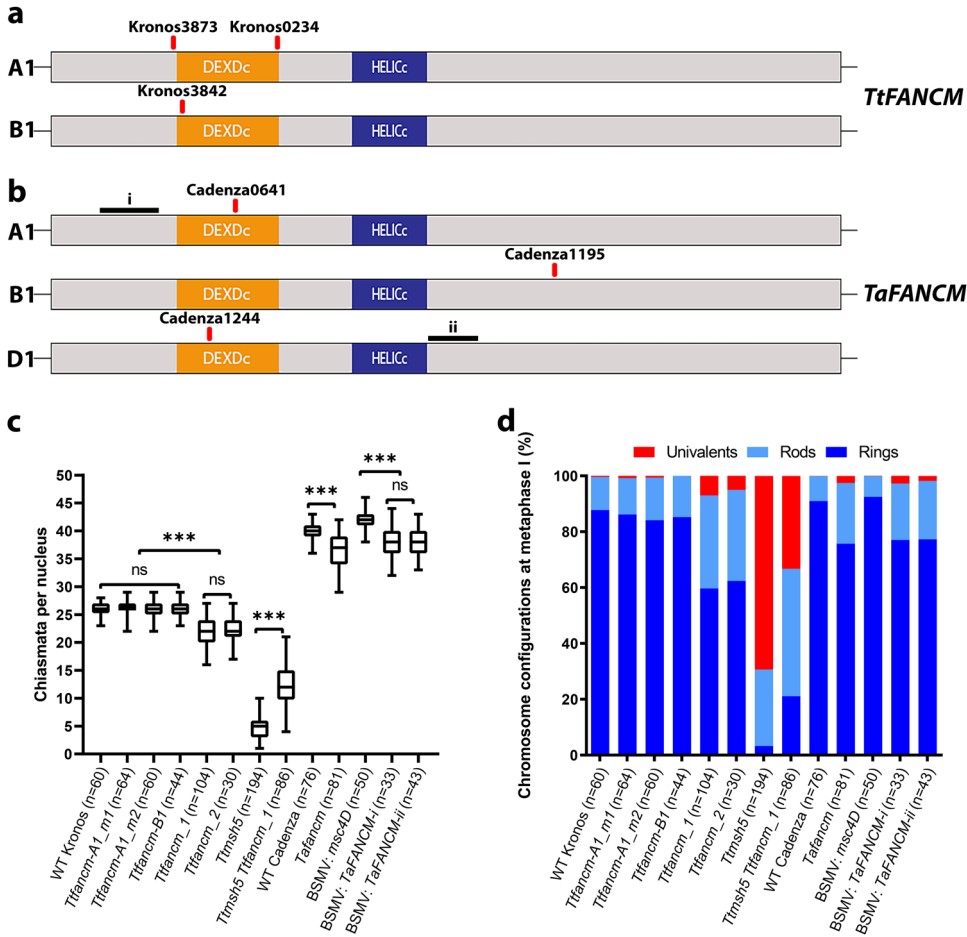

**Fig. 1 FANCM promotes formation of obligate chiasma. a** *TtFANCM* coding region for the A and B homoelogs with conserved domains shown (DEAD-like helicases superfamily (DEXDc) and helicase superfamily c-terminal (HELICc)), and TILLING mutations (red) indicated. **b** *TaFANCM* coding region for the A, B and D homoelogs with conserved domains shown (DEXDc and HELICc), and TILLING mutations (red) and VIGS target sites (black) indicated. **c** Box plot of chiasmata frequency per male meiocyte. Box extends from the 25th to 75th percentile, the centre line is the median value and the whiskers denote min and max values. The number of meiocytes sampled for each line is shown in brackets. n.s. = $p > 0.05$. *** = $p < 0.001$ (Pairwise Wilcoxon Rank Sum Test with Bonferroni correction). **d** Proportion of rings, rods and univalents per male meiocyte. The number of meiocytes sampled for each line is shown in brackets. Source data are provided as a Source Data file.

chiasma is the most likely cause for reduced pollen viability and seed production in *fancm* null mutants.

A comparative cytological analysis was then performed on *fancm* null mutants in hexaploid wheat. In *Tafancm*, $0.53 \pm 0.09$ ($n = 81$) univalents per cell were observed, compared to $0.01 \pm 0.01$ ($n = 76$) in wild type ($p < 0.001$ Pairwise Wilcoxon Rank Sum Test), and 37% of cells possessed at least one univalent pair, compared with 1.3% in wild type (Fig. 1c,d, Supplementary Fig. 3, Supplementary Data 3). The number of rod bivalents per cell also increased from $1.9 \pm 0.16$ ($n = 76$) in wild type to $4.6 \pm 0.26$ ($n = 81$) in *Tafancm* ($p < 0.001$, Mann–Whitney U Test). The increase in frequency of univalents and rod bivalents coincided with a 9% decrease in total chiasmata per cell, from $40 \pm 0.2$ ($n = 76$) in wild type to $36 \pm 0.4$ ($n = 81$) in *Tafancm* ($p < 0.001$, Mann–Whitney U Test), indicating that the requirement for FANCM in maintaining CO assurance is similar in both tetraploid and hexaploid wheat.

As the TILLING mutants have a high density of background mutations that may affect the role of FANCM based on previous studies investigating heterozygosity[21,24], *TaFANCM*-Virus Induced Gene Silencing (VIGS) was performed on hexaploid wheat. Plants inoculated with the Barley Stripe Mosaic Virus:*TaFANCM-i* construct exhibited $0.58 \pm 0.14$ ($n = 33$) univalents per

cell, compared to $0.02 \pm 0.02$ ($n = 50$) in the empty virus control ($p < 0.001$, Pairwise Wilcoxon Rank Sum Test), and 40% of cells possessed at least one univalent pair, compared with 2% in the control (Fig. 1c,d, Supplementary Fig. 3, Supplementary Data 3). The number of rod bivalents also increased from $1.56 \pm 0.18$ ($n = 50$) in the control to $4.24 \pm 0.33$ ($n = 33$) in BSMV:*TaFANCM-i* ($p < 0.001$, Pairwise Wilcoxon Rank Sum Test). The increase in frequency of univalents and rod bivalents coincided with a 9% decrease in total chiasmata per cell, from $42 \pm 0.2$ ($n = 50$) in the control to $38 \pm 0.5$ ($n = 33$) in BSMV:*TaFANCM-i* ($p < 0.001$, Pairwise Wilcoxon Rank Sum Test). The same effect was observed in a second, independent construct (BSMV:*TaFANCM-ii*) where there was no significant difference with construct (i) at MI ($p$ values $> 0.05$, Pairwise Wilcoxon Rank Sum Tests), indicating that the VIGS and TILLING mutants reproduced the same phenotype for chiasma formation.

The distribution of remaining chiasmata in *fancm* null mutants, while still deviating significantly from a Poisson distribution ($\chi^2_{(26)} = 38.9$, $n = 104$, $p < 0.05$; Fig. 3b), spanned a greater range than wild type (23–28 WT vs 16–27 *Ttfancm_1*), thus implying a defect in CO control. Therefore, an analysis utilizing pSc119.2-2, $5S$ rDNA and $45S$ rDNA synthetic oligonucleotide probes was performed on *Ttfancm_2* MI

chromosomes to determine if there was a pattern in the reduction of chiasmata. Chromosomes 1B and 6B possess 45 $S$ rDNA sites that form the Nucleolar Organization Regions (NOR) and can therefore be reliably identified[7,35] (Supplementary Fig. 5). Chromosomes 1B and 6B exhibited a significant reduction in the number of chiasmata per chromosome in $Ttfancm\_2$ compared to wild type (($1.7 \pm 0.07$, $n = 45$ to $1.3 \pm 0.11$, $n = 40$ on 1B ($p < 0.01$, Mann–Whitney U Test) and $1.8 \pm 0.06$, $n = 45$ to $1.3 \pm 0.11$, $n = 40$ on 6B ($p < 0.001$, Mann–Whitney U Test; Supplementary Fig. 5)). The Fluorescence in situ hybridization (FISH) analysis also revealed a chromosomal bias in chiasmata distribution in the wild type, as chromosomes 1B and 6B were overrepresented among rod bivalents. They accounted for 27%, indicating that chromosomes with NORs were more likely to form a single chiasma than other chromosomes ($\chi^2_{(2)} = 13.16$, $p < 0.01$). Chromosomes 1B and 6B were also overrepresented among the univalent pairs in the $Ttfancm\_2$ null mutant ($\chi^2_{(2)} = 21.07$, $p < 0.01$), accounting for 44% of all observed. However, the overall reduction in chiasmata in $Ttfancm\_2$ occurred throughout all chromosomes and 1B and 6B did not significantly deviate from expected values ($\chi^2_{(2)} = 5.83$, $p > 0.05$). Therefore, a direct consequence of the $fancm$ null mutant appears to be a reduced ability of the NOR chromosomes to buffer against a general loss of chiasmata.

**FANCM is required for localization of HEI10 at late prophase I.** As CO assurance was abrogated in the $fancm$ null mutants, localization dynamics of the class I CO protein HEI10 were investigated. HEI10 initially localizes as small, numerous, axis-associated foci during leptotene in Arabidopsis and wheat, that coarsen into fewer, larger foci, that mark class I CO sites at late prophase I[29,36,37]. At leptotene, the number of HEI10 foci per meiocyte was not significantly different between $Ttfancm\_1$ ($356 \pm 32$, $n = 7$) and wild type ($351 \pm 25$, $n = 7$) ($p > 0.05$, Mann–Whitney U Test) (Supplementary Fig. 6). At zygotene, the number of HEI10 foci was not significantly different between $Ttfancm\_1$ ($156 \pm 14$, $n = 7$) and wild type ($156 \pm 13$, $n = 9$) ($p > 0.05$, Mann–Whitney U Test) (Supplementary Fig. 6). However, at pachytene HEI10 foci decreased by 20% from $39 \pm 0.8$ ($n = 82$) in wild type to $32 \pm 0.5$ ($n = 74$) in $Ttfancm\_1$ (Mann–Whitney U Test, $p < 0.001$), (Fig. 2). Furthermore, at diakinesis HEI10 foci reduced by 29%, from $34 \pm 0.5$ ($n = 54$) in wild type to $24 \pm 0.5$ ($n = 60$) in $Ttfancm\_1$ (Mann–Whitney U Test, $p < 0.001$) (Fig. 2) and this also occurred in hexaploid wheat. At pachytene the number of HEI10 foci per meiocyte decreased by 9%, from $58 \pm 0.6$ ($n = 54$) in wild type to $53 \pm 1.2$ ($n = 34$) in the $Tafancm$ null ($p < 0.001$, Mann–Whitney U Test; Fig. 2). This reduction in HEI10 foci at pachytene suggests that formation of a proportion of class I COs is sensitive to the loss of FANCM in wheat.

**DSBs, meiotic progression, and SC formation appear normal in $fancm$.** As a proxy marker for DSB formation during meiosis, localization of the strand-exchange protein RAD51[38] was scored during leptotene. The number of RAD51 foci was not significantly different between $Ttfancm\_1$ ($1397 \pm 21$, $n = 5$) and wild type ($1403 \pm 24$, $n = 5$) ($p > 0.05$, Mann–Whitney U Test) (Supplementary Fig. 7), indicating no apparent effect on the number of early recombination sites. MSH5, an early class I CO recombination protein[7,39] localized to unsynapsed and synapsed chromosomes at zygotene, with no significant difference in the number of foci between $Ttfancm\_1$ ($336 \pm 14$ $n = 5$) and wild type ($378 \pm 22$, $n = 5$) ($p > 0.05$, Mann–Whitney U Test) (Supplementary Fig. 8).

Fixed anthers of equivalent sizes from wild type and $Ttfancm\_1$ corresponded to the same meiotic stages, indicating that there

was no delay in meiotic progression. In both cases, anthers 0.6–0.7 mm in length contained cells at leptotene, 0.8 mm at zygotene, 0.9 mm at pachytene, 0.95–1.1 mm at MI, 1.1 mm at dyad, and 1.3 mm at tetrad. Furthermore, immunolocalisation of ASYNAPSIS1 (ASY1)[40] and ZYP1[41] revealed normal axis formation and synapsis in $Ttfancm\_1$, as previously described for wild type[7,29,42]. In wild type and $Ttfancm\_1$, ASY1 formed a linear signal on the unsynapsed chromosome axes at leptotene, which depleted during zygotene on synapsed chromosomes (Supplementary Fig. 9). ZYP1 was initially detected during late-leptotene as presynaptic foci that extended throughout zygotene until a complete linear signal was observed along synapsed chromosomes at pachytene. These data indicate that early stages of meiotic recombination are unperturbed in the $fancm$ null mutants.

**FANCM limits class II crossovers.** $TtMSH5B$ possesses a natural loss-of-function deletion mutation and therefore the single $Ttmsh5a$ mutant produces a null allele of MutSγ that is defective in class I COs[7]. The $Ttmsh5$ null was crossed with the $Ttfancm\_1$ null to create the quadruple null $Ttmsh5$ $Ttfancm$ to investigate whether $fancm$ mutants can compensate for loss of class I COs. In $Ttmsh5$, $4.29 \pm 0.13$ ($n = 194$) bivalents per cell were observed, which increased ~2.2-fold to $9.34 \pm 0.24$ ($n = 86$) in $Ttmsh5$ $Ttfancm$ ($p < 0.001$, Pairwise Wilcoxon Rank Sum Test). Furthermore, ring bivalents accounted for only 3.2% of chromosome pairs in $Ttmsh5$, which significantly increased to 21% in $Ttmsh5$ $Ttfancm$ ($p < 0.001$, Pairwise Wilcoxon Rank Sum Test). The increase in ring bivalents coincided with a ~2.6-fold increase in the number of chiasmata per cell, from $4.7 \pm 0.15$ ($n = 194$) in $Ttmsh5$ to $12.4 \pm 0.41$ ($n = 86$) in $Ttmsh5$ $Ttfancm$ ($p < 0.001$, Pairwise Wilcoxon Rank Sum Test; Fig. 1c,d, Supplementary Fig. 3, Supplementary Data 3). The additional chiasmata in $Ttmsh5$ $Ttfancm$ did not deviate significantly from a Poisson-predicted distribution ($\chi^2_{(20)} = 9.74$, $n = 86$, $p > 0.05$; Fig. 3d), implying that they formed via the interference-insensitive class II CO pathway.

**FANCM localizes to meiotic chromosomes during prophase I.** To gain further insight into the role of FANCM during meiotic recombination, a wheat FANCM antibody was generated and its specificity verified by immunolocalisation. FANCM was first detected at late-leptotene as numerous ($991 \pm 42$, $n = 5$), small ($0.4$ μm $\pm 0.02$, $n = 50$), axis-associated foci (Fig. 4a). By early-zygotene fewer ($20 \pm 1.3$, $n = 22$), but larger ($0.7$ μm $\pm 0.02$, $n = 50$) distinct FANCM foci were observed of which 58% (254/441) associated with ASY1 and 42% (187/441) with ZYP1 (Fig. 4b), indicating localization to synapsed and unsynapsed chromosomes. By mid-zygotene, the number of large FANCM foci peaked at $29 \pm 1.5$ ($n = 29$), which is similar to the number of designated class I CO sites[7] (Fig. 4c). FANCM foci then decreased to $18 \pm 1.8$ ($n = 11$) at late-zygotene (Fig. 4d), and then $12 \pm 1.0$ ($n = 12$) by pachytene (Fig. 4e). When co-immunostained with anti-TaASY1, anti-TaFANCM also gave a linear signal to unsynapsed chromosomes, which was not present when used exclusively or in combination with other antibodies (e.g., anti-HvHEI10, anti-AtZYP1), indicating that cross-reactivity with ASY1 may be responsible for the linear signal. Furthermore, the linear signal was present in the $Ttfancm\_1$ null mutant, whereas the FANCM foci were absent (Fig. 4f).

As the number of zygotene FANCM foci correlated with the number of designated class I CO sites, FANCM was co-immunostained with HEI10. Due to difficulties staging the meiocytes without ASY1, an alternative anti-HvHEI10 was used which produces a linear signal on unsynapsed chromosomes and forms prominent foci on synapsed chromosomes[7]. As the HEI10

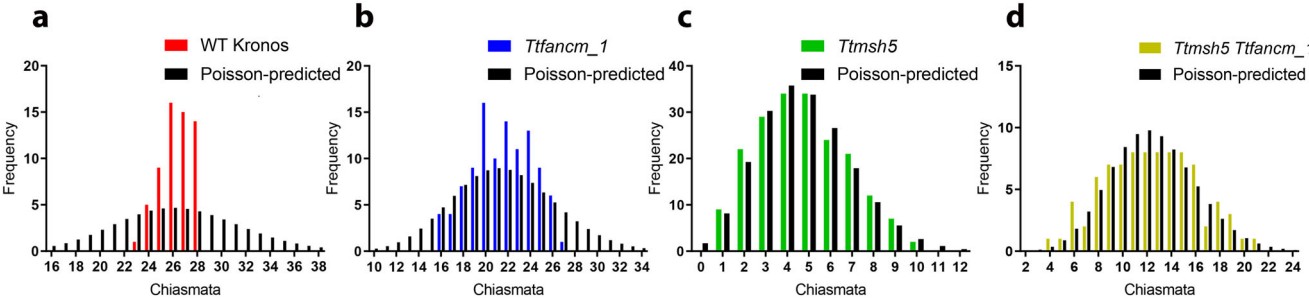

**Fig. 2 Class I crossover recombination protein HEI10 is reduced in *fancm* null mutants at late prophase I. a–f** Co-immunofluorescence of HEI10 (white) and ZYP1 (red) on meiotic prophase I chromosome spreads. Representative micrographs are shown from replicates, (**a**) (*n* = 82), (**b**) (*n* = 74), (**c**) (*n* = 54), (**d**) (*n* = 60), (**e**) (*n* = 54), and (**f**) (*n* = 34). Pachytene (**a**, **c**, **e** and **f**) and early diakinesis (**b** and **d**). Scale bars = 10 μm. **g** Counts of HEI10 foci per cell with mean values ± SD. The number of meiocytes sampled for each line is shown in brackets. *** = *p* < 0.001 (Mann–Whitney U Test). Source data are provided as a Source Data file.

**Fig. 3 FANCM limits class II crossovers. a–d** Observed and Poisson-predicted distributions of chiasma frequency per cell. **a** The observed wild-type distribution of chiasmata deviates significantly from a Poisson-predicted distribution ($\chi^2_{(28)}$ = 113.16, *n* = 60, *p* < 0.01, Chi-squared test). **b** The distribution of chiasmata in *Ttfancm_1* deviates significantly from a Poisson-predicted distribution ($\chi^2_{(26)}$ = 38.9, *n* = 104, *p* < 0.05, Chi-squared test). **c** The distribution of chiasmata in *Ttmsh5* does not deviate significantly from a Poisson-predicted distribution ($\chi^2_{(14)}$ = 5.63, *n* = 194, *p* ≥ 0.975, Chi-squared test). **d** The distribution of chiasmata in *Ttmsh5 Ttfancm_1* does not deviate significantly from a Poisson-predicted distribution ($\chi^2_{(20)}$ = 9.74, *n* = 86, *p* > 0.95, Chi-squared test). Source data are provided as a Source Data file.

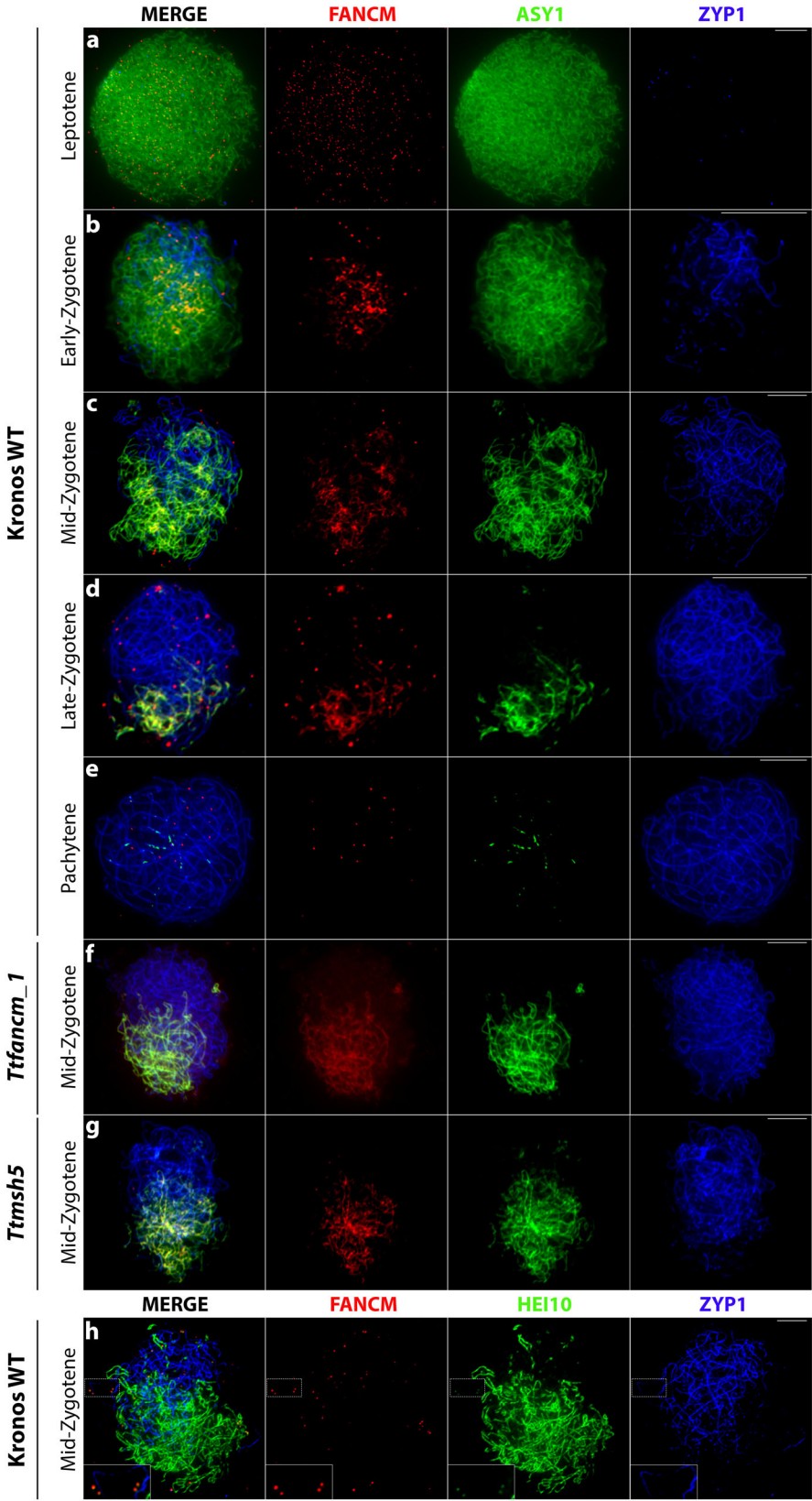

**Fig. 4 FANCM localizes to meiotic chromosomes at prophase I. a–g** Co-immunofluorescence of FANCM (red), ASY1 (green) and ZYP1 (blue) on meiotic prophase I chromosome spreads. **h** Co-immunofluorescence of FANCM (red), HEI10 (green) and ZYP1 (blue) on meiotic chromosome spreads at mid-zygotene. Representative micrographs are shown from replicates, (**a**) ($n = 5$), (**b**) ($n = 22$), (**c**) ($n = 29$), (**d**) ($n = 11$), (**e**) ($n = 12$), (**f**) ($n = 16$), (**g**) ($n = 20$), (**h**) ($n = 5$). Insets, 2× magnified view of outlined region, showing co-localization of FANCM and HEI10 foci on synapsed chromosomes. Scale bars = 10 μm.

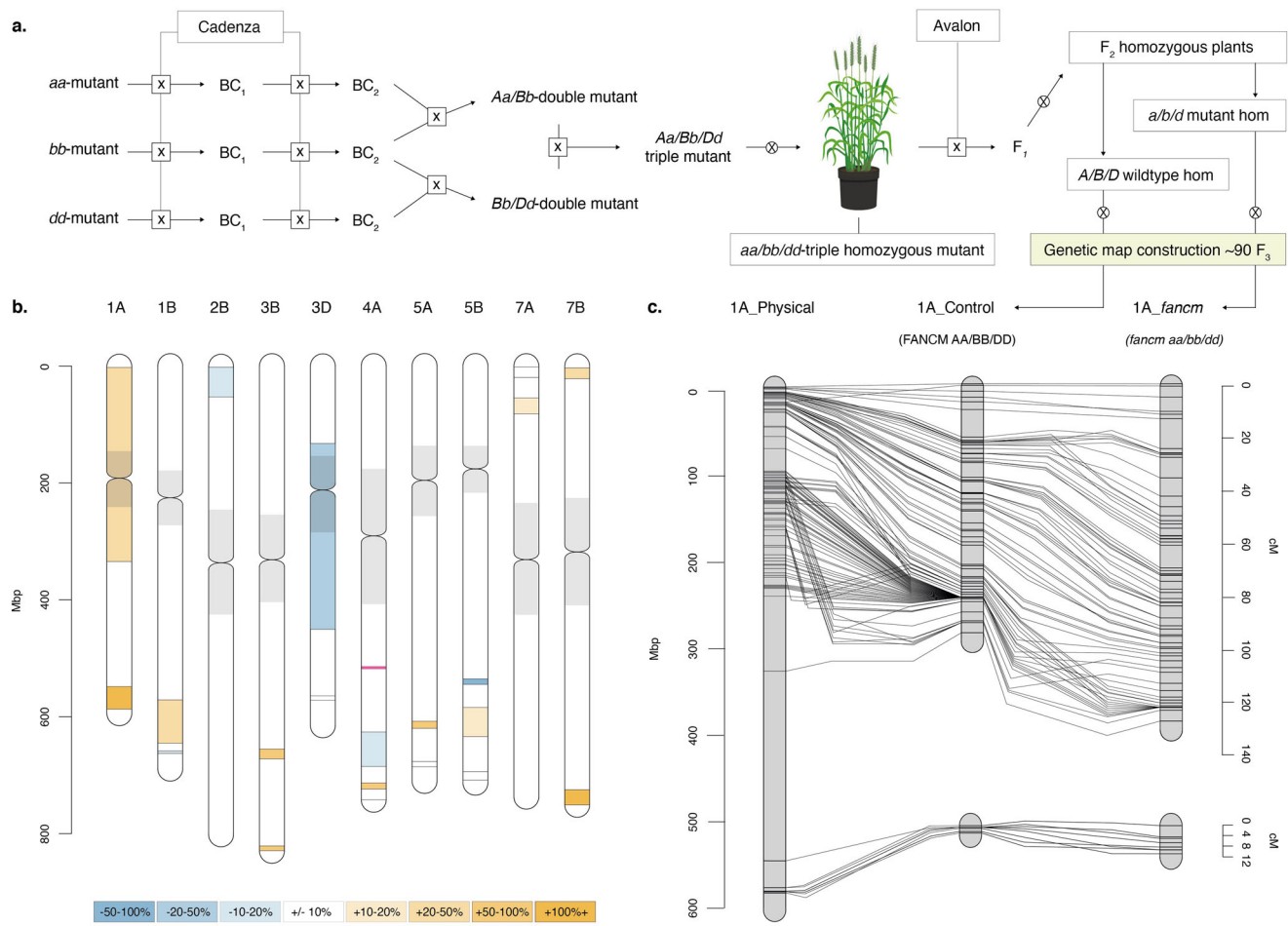

**Fig. 5 Definition of the hexaploid marker-based recombination analysis highlighting Chromosome 1 A as an example. a** Crossing schematic for the development of the triple homozygous mutant and F₃ mapping populations fixed at *fancm* for either wild type or mutant alleles. **b** Segregating regions in both F₃ populations aligned and ordered by RefSeqv1.0 coordinates (Mbp), flanking markers indicated by lines. Increasing regions (>10%) are highlighted with shades of orange and decreasing regions (<−10%) in blue, with neutral regions (−10 to +10%) in white. The location of *fancm* on chromosome 4 A is indicated by a red line and centromeric regions are shaded in gray. **c** Comparison of the genetic maps of the F₃ populations for chromosome 1 A (cM). The control population is aligned to the physical positions (Mbp) to illustrate chromosomal location.

foci on unsynapsed chromosomes were obscured by the non-specific linear signal, only FANCM foci localizing on synapsed chromosomes were scored. A mean of $14 \pm 2.7$ ($n = 5$) FANCM foci per cell were observed of which $8 \pm 2.2$ (59%) overlapped with emergent HEI10 foci (Fig. 4h). Furthermore, FANCM immunolocalisation to class I-deficient mutant *Ttmsh5* displayed a 51% reduction in the number of FANCM foci at mid-zygotene, from $29 \pm 1.5$ ($n = 29$) in wild type to $15 \pm 0.8$ ($n = 20$) in *Ttmsh5* (Fig. 4g). Taken together, these data are consistent with 51–59% FANCM foci being present at designated class I CO sites and promoting class I interfering COs, while 41–49% FANCM foci are located at potential NCO sites.

**Genetic mapping reveals an overall increase in crossovers in *fancm* mutants**. Genetic mapping using molecular marker analysis was performed on segregating tetraploid and hexaploid *fancm* null mutants as an additional tool to measure recombination frequency and distribution. The hexaploid wheat Cadenza *fancm* null mutant was crossed with Avalon to produce F₂ plants and the F₃ segregants were used for genetic map construction (Fig. 5a). Segregating Cadenza/Avalon genetic markers on the 35 K Axiom chip provided 21 intervals on ten chromosomes that were tested for recombination using a LOD 8 threshold. This revealed an increase in COs (>10%) in 11 intervals, a reduction in

five (>−10%) and no difference in 4 (between 10% and −10%) (Table 1, Fig. 5b, Supplementary Data 4, 5 & 6). Chromosome 1 A exhibited the largest increase in genetic map length in the *fancm* null mutant and is shown as an example (Fig. 5c). The map length was increased by 19.1% in *fancm* over WT when adding the cM distances across all regions, or by 31.3% when averaging the percentage cM increase from the 21 intervals.

Recombination in tetraploid wheat *fancm* mutants was measured with KASP markers designed against mapped segregating SNPs from the mutation donor lines. Four F₃ populations, two homozygous WT at *FANCM* and two fixed homozygous for the mutations, consisting of 95 plants, were tested across seven genetic intervals on 5 chromosomes. Recombination was increased in the chromosome 1B interval but decreased within the 2 A, 2B and 3 A intervals, but were the same in another 3 A interval and 3B (Table 2, Supplementary Data 7). The trend was an overall decrease, but statistically the mutant and wild type were indistinguishable in genetic map length for the intervals tested.

**Crossover rate differences in *fancm* correlate wild type crossover distributions**. Genomic regional variation in CO rate differences between *fancm* and wild type shows a significant positive correlation with a previously published wild type CO rate map

**Table 1 Recombination analysis in the hexaploid Avalon x Cadenza F₃ *fancm* mutant mapping population.**

| Chr. | Markers | Marker region | Position | Control | *fancm* | Delta | % Difference |
|---|---|---|---|---|---|---|---|
| 1A_1 | 162 | AX-94733072 - AX-94752690 | 2378195–332208434 | 93.76 | 126.60 | 32.84 | 35.0% |
| 1A_2 | 12 | AX-94874424 - AX-95151237 | 548494535–586914258 | 3.33 | 11.44 | 8.11 | 243.3% |
| 1B_1 | 116 | AX-95025932 - AX-94731046 | 571060391–645252540 | 67.91 | 83.51 | 15.61 | 23.0% |
| 1B_2 | 13 | AX-94813152 - AX-94804524 | 658531550–662842760 | 12.24 | 10.66 | −1.58 | −12.9% |
| 2B | 63 | AX-95630341 - AX-94643790 | 2177758–52907517 | 46.89 | 40.27 | −6.62 | −14.1% |
| 3B_1 | 19 | AX-94653790 - AX-94633883 | 655306481–672099880 | 11.79 | 18.27 | 6.48 | 55.0% |
| 3B_2 | 17 | AX-95248765 - AX-94990660 | 821077029–829286625 | 7.34 | 11.99 | 4.65 | 63.3% |
| 3D_1 | 15 | AX-94534357 - AX-94667914 | 132604563–450294132 | 37.08 | 27.06 | −10.02 | −27.0% |
| 3D_2 | 9 | AX-95010529 - AX-94721213 | 564307274–571493665 | 19.87 | 20.92 | 1.05 | 5.3% |
| 4A_1 | 10 | AX-95159296 - AX-94999060 | 626119038–684968665 | 14.14 | 12.34 | −1.80 | −12.7% |
| 4A_2 | 4 | AX-94775503 - AX-94877844 | 713351841–723516355 | 10.52 | 20.58 | 10.06 | 95.6% |
| 4A_3 | 32 | AX-95153315 - AX-94555122 | 726214864–742002004 | 28.21 | 29.18 | 0.97 | 3.4% |
| 5A_1 | 25 | AX-94654340 - AX-95257149 | 607681540–619778700 | 5.34 | 9.57 | 4.24 | 79.4% |
| 5A_2 | 24 | AX-94995722 - AX-95230303 | 676592388–685543181 | 10.22 | 9.89 | −0.33 | −3.2% |
| 5B_1 | 13 | AX-95106649 - AX-95258242 | 535359886–544608954 | 20.53 | 4.51 | −16.02 | −78.0% |
| 5B_2 | 74 | AX-95151783 - AX-94657489 | 584127473–633943398 | 20.12 | 23.19 | 3.08 | 15.3% |
| 5B_3 | 14 | AX-94885467 - AX-94474369 | 693674319–708229373 | 18.74 | 20.17 | 1.43 | 7.6% |
| 7A_1 | 48 | AX-94485699 - AX-94664110 | 1704442–20294508 | 36.84 | 38.44 | 1.60 | 4.4% |
| 7A_2 | 30 | AX-94802515 - AX-94505773 | 61887949–83631459 | 22.92 | 26.30 | 3.38 | 14.8% |
| 7B_1 | 22 | AX-94733039 - AX-95191125 | 3338679–21668399 | 26.66 | 32.04 | 5.38 | 20.2% |
| 7B_2 | 48 | AX-94582074 - AX-95104628 | 724954096–750605855 | 29.68 | 70.99 | 41.31 | 139.2% |
| | | | Average all intervals | | | 4.94 | 31.3% |
| | | | Total map length (cM) | 544.13 | 647.92 | 103.79 | 19.1% |

**Table 2 Recombination analysis in the tetraploid Kronos F₃ *fancm* mutant mapping population.**

| Chr. | Markers | Marker region | Position | Control | *fancm* | Delta | % Difference |
|---|---|---|---|---|---|---|---|
| 1B | 2 | K3842.1B.563292311-K3842.1B.591113626 | 563292311–591113626 | 10.3 | 13.6 | 3.3 | 32.00% |
| 2A | 2 | K3842.2A.379735839-K3842.2A.651081672 | 379735839–651081672 | 21.8 | 16.3 | −5.5 | −25.40% |
| 3A | 2 | K3842.3A.507462812-K3842.3A.692623524 | 507462812–692623524 | 27.7 | 30 | 2.3 | 8.20% |
| 2B | 2 | K3873.2B.223733552-K3873.2B.486585051 | 223733552–486585051 | 10 | 6.5 | −3.5 | −34.80% |
| 3B | 3 | K3873.3B.223037264-K3873.3B.723453707 | 223037264–723453707 | 24.7 | 26.6 | 1.9 | 7.80% |
| 2B | 2 | K3842.2B.367667794-K3842.2B.545253995 | 367667794–545253995 | 9.9 | 8.8 | −1 | −10.60% |
| 3A | 2 | K3873.3A.53209641-K3873.3A.423481141 | 53209641–423481141 | 13.5 | 11.5 | −1.9 | −14.30% |
| | | | Average all intervals | | | −0.63 | −5.30% |
| | | | Total map length (cM) | 117.9 | 113.3 | −4.6 | −3.9% |

derived from a Chinese Spring × Renan recombinant inbred population[43] (Supplementary Fig. 10). Differential CO rate in *fancm* also shows non-significant positive relationships with ChIP-seq signals for the chromosome axis protein ASY1 and the meiotic recombinase DMC1, the euchromatic marks H3K4me1, H3K4me3 and H3K27ac, and the Polycomb mark H3K27me3, and non-significant negative correlations with H3K9me2, H3K27me1, centromeric CENH3, and DNA methylation in CG and CHG contexts (Supplementary Fig. 10, Supplementary Data 8 and 9). Therefore, while COs are increased in *fancm*, their genomic distribution is comparable to that in wild type. This suggests that chromatin may influence the recombination landscape in similar ways in both wild type and *fancm*.

## Discussion
In the absence of FANCM, 64% of metaphase I cells in the tetraploid and 37% in the hexaploid possessed at least one univalent pair, thus indicating loss of CO assurance. This led to chromosome mis-segregation during meiosis II, followed by a decrease in pollen viability and seed production. The metaphase I FISH analysis revealed an overall reduction in chiasmata, suggesting a random loss of COs, consistent with studies in lettuce *fancm* mutants[25]. Nucleolar Organizing Region containing chromosomes 1B and 6B exhibited a lower level of chiasmata in wild type and exhibited univalents more frequently in *fancm* than other chromosomes. The loss of the obligate chiasma and reduction of HEI10 foci from 34 to 24 during diakinesis in tetraploid wheat suggested that loss of FANCM had a direct effect on the class CO I pathway. These data are consistent with the increase of rod bivalents observed in *A. thaliana fancm* mutants[20], and increased numbers of univalents observed in lettuce *fancm*[25].

FANCM localized as small, numerous foci during leptotene that coarsened into fewer, large foci during zygotene that correlated with the number of expected COs. This pattern is consistent with FANCM localizing to influence the fate of a proportion of recombination intermediates. FANCM could be required to release resected DSB ends from invasion of sister chromatids to enable engagement with the homolog, as reported in budding yeast[44]. However, reduced inter-homolog engagement of the first strand may not have a significant effect on CO formation in wheat, due to the ~50:1 excess of DSBs over COs[14], which may mask this effect, compared to budding yeast where the ratio of DSBs to COs is ~1.7:1[45]. This effect could be tested by crossing wheat *fancm* and *spo11* hypomorphic mutants to systematically reduce the number of DSBs, thus enabling comparison of inter-homolog engagement in *spo11* and *fancm/spo11* mutants. If there was an early role for FANCM a greater loss of inter-homolog engagement would be expected in the *fancm/spo11* mutants compared to *spo11*.

In tetraploid wheat, ~29 FANCM foci coarsened during zygotene of which ~51–59% localized with HEI10. In the absence of FANCM, HEI10 foci reduced from 34 to 24 per nucleus at diakinesis, suggesting that FANCM is required to promote 29% of class I COs. Our data are consistent with a model in which FANCM facilitates second-end capture at class I CO sites. FANCM may release the DSB second-end from association with the sister-chromatid, or other homologous/homoeologous templates, similarly to budding yeast, but at a later stage. This model is consistent with current data from rice where FANCM interacts with RPA1a during meiosis to promote CO formation[46], and in Arabidopsis RPA1a plays a crucial role in second-end capture[47]. Our data supports the molecular function of FANCM as a DNA helicase that unwinds strand invasion D-loop intermediates during zygotene. The requirement for FANCM to function at a subset of designated class I CO sites may be due to a stochastic propensity for DSB second-ends to invade the sister chromatid, rather than the homolog. This would also imply that CO site designation may occur at the D-loop stage in wheat.

A role for FANCM in limiting the class II CO pathway was revealed by crossing the mutant with the class I CO mutant msh5[7]. Chiasmata increased over ~2.6-fold in the fancm/msh5 quadruple mutant compared to the msh5 mutant, although this was insufficient to restore CO levels, as reported in Arabidopsis[19,20]. Therefore, FANCM may unwind a small proportion of D-loop intermediates between homologous chromosomes that were destined to form NCOs, but in its absence may be processed by the class II CO pathway.

These data suggest that FANCM reduces the probability of non-interfering class II CO's from forming, whilst promoting interfering class I CO's, thus producing a landscape where COs are spaced further apart. FANCM appears to ensure the fate of a subset of recombination intermediates, rather than being involved in CO interference per se, but a recent analysis of the role of FANCM and its interacting partner FANCD2 in Arabidopsis suggests that it directly promotes interference of class I COs, based on HEI10 foci frequency and distribution[25]. However, due to the significant reduction of HEI10 foci in wheat, this effect may be obscured, but still active.

The data presented here suggest that FANCM is required at CO designated sites/NCOs during zygotene. In wild type tetraploid wheat, the ~10-fold excess of HEI10 foci at mid-zygotene (351) compared to the eventual (34) foci at diakinesis superficially should be sufficient to enable CO assurance. However, in the absence of the 29 FANCM foci (of which ~51–59% localized with HEI10), the eventual HEI10 foci reduced to 24 at diakinesis in the mutant. Therefore, these data are consistent with class I CO designation occurring before FANCM coarsens at mid-zygotene, as the majority of the remaining HEI10 were unable to compensate. Recent studies have shown that CO interference is mediated by ZYP1 in Arabidopsis[17,18], although modelling of CO designation with the coarsening of HEI10 foci occurring during mid/late prophase I may also play a role[37]. The FANCM data here indicates that CO designation occurs earlier than mid-zygotene, consistent with budding yeast[3]. This also raises the question of how FANCM is regulated to function at both designated class I CO sites and non-designated sites to ensure the optimum fate of recombination intermediates.

The wheat fancm mutants exhibited a reduction in fertility (36% in tetraploid and 15% in hexaploid wheat), due to loss of the obligate chiasma. This is consistent with a previous study utilizing FANCM-VIGS on tetraploid wheat $F_1$ hybrids that showed a significant reduction in fertility, but no differences in recombination on chromosome 1A[48]. The immunohistochemistry, cytological and molecular marker analysis presented here reveal that loss of class I COs are offset by an increase in class II COs in the

tetraploid, thus resulting in no net change. However, fewer HEI10 foci were reduced in hexaploid wheat fancm mutants, compared to the tetraploid, leading to an overall increase of COs by 31%. Based on an increase of class II chiasmata detected in the tetraploid, it is likely that the increase in COs in the hexaploid arose via the class II CO pathway. The molecular marker data revealed that COs increased in the majority of intervals tested, thus providing an opportunity to modulate recombination in wheat breeding programs. The distal H3K27me3-enriched regions experienced the greatest increase in COs, whereas the interstitial/proximal regions were more likely to be lower, similar to recombination increases in the barley recq4 mutant[49].

## Methods

**Identification of wheat FANCM.** Wheat FANCM orthologues were identified using the Arabidopsis thaliana amino acid sequences (At1g35530) to BLAST against publicly available databases: Triticum turgidum[50] Svevo.v1 https://plants.ensembl.org/Triticum_turgidum) and T. aestivum[43] IWGSC https://plants.ensembl.org/Triticum_aestivum). Wheat cds were aligned using the Clustal W algorithm (gap open cost = 12, gap extend cost = 3), translated and the primary sequence ran through the Conserved Domain Database (NCBI).

**Plant material.** Triticum turgidum 'Kronos' was used as a wild-type control for experiments involving the tetraploid Kronos TILLING mutant lines obtained from www.SeedStor.ac.uk[51]. We selected knockout mutants (premature termination codon; Supplementary Data 1) in TtFANCM-A1 (TRITD4Av1G171480) exon 2 (Kronos3873; abbreviated K3873 hereafter) and exon 8 (K0234). Similarly, we selected a knockout mutant (K3842) in the third exon of TtFANCM-B1 (TRITD4Bv1G035000). We confirmed the mutations via genotyping (Supplementary Data 2) and named the single mutants as Ttfancm-A1_m1 (K3873), Ttfancm-A1_m2 (K0234), and Ttfancm-B1 (K3842). We also generated double null mutants: Ttfancm_1 (K3873 x K3842) and Ttfancm_2 (K0234 x K3842) and characterized a Ttmsh5-A1 mutant (K863), which is functionally a null (Ttmsh5) as there is a natural loss-of-function deletion in TtMSH5-B1 (Desjardins et al. 2020), and a Ttfancm Ttmsh5 double mutant (K3873 × K3842 × K863). Triticum aestivum cv. 'Cadenza' was used as a wild-type control for experiments involving hexaploid Cadenza TILLING mutant lines. We selected knockout mutants in all three TaFANCM homoeologs including TaFANCM-A1 (TraesCS4A02G217700; Cadenza0641, abbreviated C0641 hereafter), TaFANCM-B1 (TraesCS4B02G096400; C1195) and TaFANCM-D1 (TraesCS4D02G092800; C0827). The single mutants were crossed to generate a Tafancm null mutant (C0641 × C1195 × C0827)[52]. Plants were grown under controlled environmental growth conditions: photoperiod 16 h, temperature 21 °C (day)/16 °C (night) and relative humidity ~60%. VIGS experiments were conducted in a Level 3 biological containment facility at Rothamsted Research.

**RT-PCR.** Total RNA was extracted from T. turgidum cv. 'Kronos' and T. aestivum cv. 'Cadenza' spikes using ISOLATE II RNA Mini Kit (Bioline), and cDNA was synthesized with the Tetro cDNA synthesis kit (Bioline). The coding sequences for both varieties were amplified and sequenced using homoeolog-specific primers: FANCM_A_F 5′-CTGGATGTTGGCTGCACTCG-3′ and FANCM_A_R 5′-ATGTGGTTGCTTTCAGAGGTA-3′, FANCM_B_F 5′-AGAGGCTATGTTTC-TATCACC-3′ and FANCM_B_R 5′-GATCCTGATGTATTCCCTACC-3′, FANCM_D_F 5′-AAGGAAGGAAGTGGGAAAGT-3′ and FANCM_D_R 5′-CCAGGCAAGCATGAATATCC-3′.

**Virus induced gene silencing.** The barley stripe mosaic virus, virus induced gene silencing system (BSMV:VIGS)[53,54] was used to target expression of TaFANCM in T. aestivum 'Bobwhite'. Two non-overlapping regions with high predicted silencing efficiency were selected, following an in silico analysis of TaFANCM-A1 cds (TraesCS4A02G217700) with si-Fi21 (siRNA Finder). For the first selected region, primers TaFANCM-vigs-i_F 5′-**AAGGAAGTTTAA**GTGGACTCATTCACCAG-GAG-3′ and TaFANCM-vigs-i_R 5′-**AACCACCACCACCGT**TCATCGCCACG-GACAGCA-3′ were used to amplify 259–572 bp of TraesCS4A02G217700 coding region with Q5 DNA polymerase (NEB). For the second region, primers TaFANCM-vigs-ii_F 5′-**AAGGAAGTTTAA**GTGAAGGACCAGAGCTGCAAG-3′ and TaFANCM-vigs-ii_R 5′-AACCACCACCACCGTGCCTGAACTGAAG-TACTGAGC-3′ were used to amplify 2102–2376 bp of TraesCS4D02G092800 coding region with Q5 DNA polymerase (NEB). The amplicons were cloned into pCa-ybLIC to create a recombinant BSMV RNAγ. The recombinant pCa-ybLIC, as well pCaBS-α and pCaBS-β, were then transformed into electrocompetent Agrobacterium tumefaciens (GV3101), agroinfiltrated, and the virus accumulated in intermediate host Nicotiana benthamiana, prior to inoculation of wheat plants with infected sap at the 4–4.5 leaf stage (~28 days post-sowing)"[54]. For cytological analysis, anthers were dissected and fixed in 3:1 (v/v) ethanol: acetic acid at 14-days post inoculation.

**FANCM antibody production**. Primers AbF 5′-AGCA-TATGTTGGTGGCAGCTGGTGTG-3′ and AbR 5′-TCCTCGA-GATTGTATTCCCCAGCTCG-3′ were used to amplify 1107–1935 bp of the *TraesCS4B02G096400* (*FANCM-B1*) coding region with Q5 DNA polymerase (NEB). The PCR product was ligated into pDrive (Qiagen) and sequenced (Eurofins). pDrive was then digested with *Nde*I and *Xho*I, and the insert was ligated into pET21b (Merck-Millipore). Transformed BL21 (DE3) cells expressed a 32 kDa recombinant protein that was used to produce a rat anti-FANCM antibody (DC Biosciences).

**Fertility assessment**. Fertility was assessed in *Ttfancm_1* and *Tafancm* null mutants by counting the total number of seeds per plant and comparing with wild type using Mann–Whitney U tests (Minitab v 18.1.0.0). The ratio of viable to non-viable pollen grains was also determined for *Ttfancm_1* using Alexander staining[55].

**Cytological procedures**. Meiotic chromosome spreads were prepared from fixed anthers. Following digestion (30–60 min at 37 °C) in citrate buffer containing 0.33% cellulase (Duchefa Biochemie) and 0.33% pectolyase (MP Biomedicals), the anthers were macerated in a drop of water, spread in 60–80% acetic acid at 45 °C, and refixed in 3:1 (v/v) ethanol: acetic acid[54,56]. Slides were mounted in VEC-TASHIELD® Mounting Medium with DAPI (Vector Laboratories). Fluorescence in situ hybridization (FISH) was conducted using three labelled synthetic oligo-nucleotides: pSc119.2-2 [A488][35], pTa794-1 (5 S) [TxRd][7] and pTa71-1 (45 S) [A647][35]. Nikon Ni-E and Eclipse Ci fluorescence microscopes equipped with NIS elements software were used to capture and quantify images. Chiasmata number was interpreted according to bivalent shape[29]. Rod bivalents were scored as a minimum of 1 and ring bivalents scored as a minimum of 2, with additional chiasmata scored based on shape.

Chromosome spreads for HEI10 immunostaining were prepared as above. For all other antibodies, chromosome spreads were prepared from fresh anthers. Following digestion (8 min at 37 °C) in enzyme mixture containing 0.4% cytohelicase from *Helix pomatia* (Sigma-Aldrich), 1.5% sucrose and 1% polyvinylpolypyrrolidone (Sigma-Aldrich), meiocytes were squeezed out of anthers using a brass rod, spread in 0.4–1% lipsol detergent (Appleton Woods Ltd) and fixed in 4% ice-cold paraformaldehyde (Fisher Scientific)[54]. Immunostaining was conducted using the following primary antibodies: anti-TaASY1 guinea pig, 1:500;[7] anti-AtZYP1C rat, 1:500;[41] anti-AtZYP1C rabbit, 1:500;[57] anti-AtRAD51 rabbit, 1:200;[58] anti-MSH5 rabbit[12], 1:200;[7] anti-HvHEI10 rabbit, 1:250[7]. Secondary antibodies that were used, 1:200: goat anti-guinea pig AMCA (Jackson ImmunoResearch; 106-155-003); goat anti-guinea pig Alexa Fluor® 488 (Abcam; #A-11073); goat anti-guinea pig Alexa Fluor® 594 (Abcam; #A-11076); goat anti-rat AMCA (Jackson ImmunoResearch; 112-155-003); goat anti-rat Alexa Fluor® 594 (Invitrogen; #A-11007); goat anti-rabbit AMCA (Jackson ImmunoResearch; 111-155-003); goat anti-rabbit Alexa Fluor® 488 (Invitrogen; #A-11008) and goat anti-rabbit DyLight® 594 (Vector Labs; DI-1595). Meiocytes were staged with anti-ZYP1 or anti-ASY1 to ensure focal counts were made at equivalent stages; for anti-RAD51 (leptotene), anti-MSH5 (zygotene) and anti-HEI10 (leptotene to diakinesis). Counts were performed using NIS software and significance established using Pairwise Wilcoxon Rank Sum with Bonferroni correction (RStudio v 1.2.5033), Mann–Whitney U (Minitab v 18.1.0.0) or Chi-squared ($\chi^2$) tests, as appropriate.

**Kronos tetraploid F₃ marker-based recombination analysis**. *Fancm* double null mutants (*Ttfancm_1*) were developed by crossing plants with premature stop mutations in the A (Kronos3873.chr4A.517401432) and B (Kronos3842.chr4B.100316098) genomes (Supplementary Data 1). F₁ plants were checked for heterozygosity with mutation-specific KASP markers (Supplementary Data 2) and self-pollinated for F₂ seed. Four F₂ sibling plants, two fixed for WT *FANCM* and two fixed for mutations in the A&B homoeologs (*Ttfancm_1*) were self-pollinated to produce F₃ seed. Genome-wide mutations from donor TILLING lines were extracted (http://www.wheat-tilling.com/[51], with regions and primers selected for screening based on chromosome position and genome specificity (Supplementary Data 2). KASP primers were tested on the F₂ plants to check for heterozygosity in fourteen selected regions, seven of which were found to be segregating in at least one WT and one *fancm* double mutant population. F₃ populations of 95 plants were sown for the four F₂ lines (two fixed for WT *FANCM* and two fixed for the A and B homoeolog *fancm* mutations). The seven regions were screened for recombination analysis with either two or three markers per region. After genotyping, the final number of lines for analysis was determined by excluding missing lines or lines with missing data, and the recombination distance calculated (recombinants/total progeny × 2) × 100.

**Avalon x Cadenza hexaploid F₃ marker-based recombination analysis**. *Fancm* triple mutants (*Tafancm*) were produced by crossing mutants carrying premature stop mutations from the three homoeologs (Cadenza0641.chr4A.517403720, Cadenza1195.chr4B.100321720 and Cadenza0827.chr4D.67793583) and tracking mutations with mutation-specific KASP markers. Triple mutant *Tafancm* homozygous plants were crossed with Avalon and the F₁ progeny self-pollinated. F₂

siblings fixed as either homozygous wild type for the three *FANCM* homoeologs, or homozygous for all three mutations at *fancm* were self-pollinated to obtain F₃ seed. F₃ populations of 90 and 91 plants respectively, were sown from populations arising from fixed WT *FANCM* and triple mutant *fancm* F₂ siblings. F₃ seed from an Avalon × Cadenza cross were germinated and grown in pots filled with peat-based soil and kept in a glasshouse at 15–18 °C with 16 h light, 8 h dark. Leaf-tissue was harvested from individual plants 14 days post-sowing, when the plants were at an early seedling stage. DNA was extracted following the protocol[59] with minor modifications. DNA concentration was assessed using a Qubit 2.0 Fluorometer and was then normalized to 23 ng/μl ready for analysis with the Axiom® Wheat Breeder's array[59]. Sample preparation for array genotyping was performed with the Beckman Coulter Biomek FX. Samples were then genotyped using the Axiom® 35 K Wheat Breeders array in conjunction with the GeneTitan® using standard Affy-metrix protocols[59]. Axiom Analysis Suite (version 3.1.51.0) was used to assign genotype calls. Axiom 35 K genotyping data was ordered by chromosome and physical position obtained from IWGSC RefSeq v.1.0 (IWGSC, 2018). Mono-morphic markers (>90% of plants carrying either the Cadenza or Avalon allele), markers with high levels of missing data (>20%), markers with distorted segre-gation (heterozygous < 25% or >75%) and dominant markers (Allele 0/2 < 5) were removed prior to mapping. The remaining segregating markers were compared for both WT and triple mutant *fancm* populations. A total of 13 regions were selected to investigate recombination frequency based on the presence of identical markers segregating in both populations. Genetic mapping was carried out for each region individually using MSTmap online (http://www.mstmap.org/) using a LOD 8 threshold. Where linkage groups contained identical markers, cM distance was compared between populations. In some cases, selected regions were split into two or more linkage groups. The 'cM map length increase' was calculated as the per-centage increase of the total *fancm* cM distance over WT control across all regions and the 'cumulative percentage increase' describes the average of the percentage difference of all chromosome regions.

**Associating FANCM crossovers with chromatin marks**. Differential CO rate (*fancm* cM/Mb minus wild type cM/Mb), wild type CO rate[43], and chromatin and meiotic protein mean signals, profiled via ChIP-seq or bisulfite-seq (as analyzed in ref. [33]), were calculated within each genetic marker interval. Spearman's rank-order correlation coefficients ($r_s$) were computed for each parameter pair and plotted as a correlation matrix, with $r_s$ denoted by cell color (R v 4.0.0). P values for $r_s$ correl-ation coefficients were standardized to represent those based on pairwise values across 100 marker intervals and are indicated within each cell of the correlation matrix. Included data sets are differential CO rate (*fancm* cM/Mb minus wild type cM/Mb; "Diff_cMMb"), wild type CO rate derived from a Chinese Spring x Renan genetic map ("IWGSC_cMMb")[43], ASY1, DMC1, H3K4me3, H3K9me2 and H3K27me1 ChIP-seq[33], H3K4me1 and H3K27ac ChIP-seq[60], H3K27me3 and H3K36me3 ChIP-seq[43], CENH3 ChIP-seq[61], whole-genome bisulfite sequencing-derived DNA methylation (mCG, mCHH and mCHG proportions)[43], and the distance between the midpoint of each marker interval and the midpoint of pre-viously defined centromeric coordinates ("Dist_to_CEN")[43]. Histone ChIP-seq and bisulfite-seq data are derived from Chinese Spring seedlings or leaf tissue[61,43,60,33], and ASY1 and DMC1 ChIP-seq data are derived from Chinese Spring immature pre-emergence spikes[33].

**Reporting summary**. Further information on research design is available in the Nature Research Reporting Summary linked to this article.

## Data availability

Data supporting the findings of this work are available within the paper and its Supplementary Information files. A reporting summary for this paper is available as a Supplementary Information file. Raw data of the images are available from the corresponding author upon request. Source data are provided with the paper. Source data are provided with this paper.

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

## Acknowledgements

We would like to thank Neelam Dave for technical support, Helen Harper as sLola project coordinator, Claire Meade for support with crosses, and the sLola Steering Committee for helpful advice throughout the project. We thank Tobin Florio at flozbox-science for illustrating Fig. 5. This work was funded by UKRI through a BBSRC strategic Long and Large grant (sLoLa) BB/N002628/1.

## Author contributions

S.D.D., J.S., I.G., A.J.B., and K.K. performed the research; J.D.H., S.D.D., J.S., C.U., and K.J.E. designed the study; J.D.H., S.D.D., A.J.T., I.R.H., J.S., C.U., and K.J.E. analyzed the data; S.D.D., J.S., A.J.T., E.S.M., F.C.H.F., I.R.H., C.U., K.J.E., and J.D.H. wrote the paper. J.D.H. agrees to serve as the author responsible for contact and ensures communication.

## Competing interests

The authors declare no competing interests.
