## [Peer Review File · Nature Communications]

FANCM promotes class I interfering crossovers and suppresses class II non-interfering crossovers in wheat meiosisReviewers' Comments:

Reviewer #1:

Remarks to the Author:

This manuscript submitted by Desjardins et al reports on a study of FANCM's functions in maintaining obligate class I crossovers and suppressing class II crossovers in wheat. By analyzing *fancm* mutants in tetraploid and hexaploid wheat, the authors report that FANCM is required for normal fertility as mutants exhibited increased univalents and reduced chiasmata. Similarly, TaFANCM-i mutants by VIGS showed comparable defects. In contrast to a narrow distribution of chiasma numbers among wild type cells, mutants display a greater range of chiasma numbers observed. By FISH analysis of rDNA on chromosomes 1B and 6B, the overall reduction of chiasmata in mutants was observed. Authors then showed decreased HEI10 foci in mutants during later stages and elevated numbers of MUS81 foci, suggesting that FANCM plays a role in ensuring class I crossovers and inhibiting class II crossovers. The inhibition of class II crossovers by FANCM was also demonstrated by quadruple mutants of *Ttmsh5 Ttfancm*. FANCM first forms numerous axis-associated foci and, by mid-zygotene, FANCM appears as larger foci that overlapped partially with HEI10 foci. In *Ttmsh5* mutants, FANCM foci were found to reduce by about 51%, which may imply that FANCM may be able to locate on sites other than class I COs. In addition, the authors showed that genetic distances were overall increased in hexaploid *fancm* mutants. Last, the authors showed the correlation between DNA methylation and histone marks with crossover rate intervals.

Overall, this study is well conducted and presented. The Introduction is generally an excellent presentation of the background. The experimental analyses are quite clearly presented. I particularly appreciated the analysis of FANCM in tetraploid and hexaploid wheat, as well as by VIGS. I also appreciate the amount of work the authors put into in genetic mapping and correlation with chromatin marks. My comments and suggestions are listed below.

1. The FISH results seem to be less informative and didn't turn into a clear conclusion. As the pSc119.2-2 probe also marks other chromosomes, is it possible to include an analysis of other chromosomes? It will be also helpful if the authors can identify and label some chromosomes in Fig S5.
2. Although there is a notable decrease (20%) in HEI10 foci in mutants at pachytene stage, it still seems ambiguous to say that FANCM directly promotes class I COs. For example, FANCM may be involved in dissociating the undesired invasions (eg. between sisters or other types of intermediates), so that fewer inter-homolog invasions can be labeled by HEI10? In addition, please provide reference to support the implication of "localization of HEI10 may represent double Holliday junctions" (lines 407-408).
3. As the authors acknowledge, the chiasma numbers may be underestimated. I am still concerned that the numbers of chiasmata are much lower than observed HEI10 foci (only for CO I). Thus, I wonder how reliable the chiasma analyses are.
4. Since one HEI10 antibody produces a linear signal on unsynapsed chromosomes, is it possible that the FANCM signals on unsynapsed signals are real? Another related question is whether the FANCM antibody can still recognize truncated proteins in *Ttfancm_1* mutant? (Fig 4F)

Reviewer #2:

Remarks to the Author:

Importance of the research

Meiotic recombination events (crossovers) are at the heart of genetic improvement as they ensure a faithful production of fertile gametes and they allow shuffling of alleles to create new innovative and

powerful combinations. However, crossovers are rare events (at least one mandatory but rarely more than three per homologous pair) and there is therefore an interest to improve the rate in crops to speed up the development of performing varieties. Several genes that increase recombination rate have been found in the model species *Arabidopsis* and it is crucial to estimate how these genes affect recombination in crops, especially in polyploids like wheat. Among these genes is the helicase FANCM (Fanconi anemia complementation group M) that shows contradictory results in *Arabidopsis*: *fancm* mutation increases recombination in inbred lines but has no effect in hybrid context. Interestingly, *fancm* mutation increases significantly recombination in Brassica, rice and pea. It is thus of main interest to see whether similar results are observed in wheat.

Noteworthy results

Impact of the mutation of *fancm* in tetraploid and hexaploid wheats is detailed. Fertility of the double and triple mutants is reduced compared to wild types and simultaneous reduced viability of pollen suggests a default during gamete formation. Double mutation of *fancm* in a tetraploid background reduces the number of chiasmata. This reduction resulted from the presence of univalents at metaphase I leading to further chromosome mis-segregation. Same result was observed in hexaploid *fancm* mutants and this was confirmed using a VIGS approach. Use of HEI10 antibody shows that type I COs are slightly reduced in *fancm* mutants at late prophase I only suggesting that at this late stage, location of HEI10 on chromosomes depends on FANCM. The number of double strand breaks as well as synapsis and meiotic progression were not affected in mutants. Implication of FANCM in class II COs was confirmed using double *msh5/fancm* mutants and FANCM was immunolocalized on chromosomes using a wheat-specific antibody. Mutation of *fancm* resulted in a significant increase of genetic distances in hexaploid context but was not changed in the tetraploid. Overall, this study provides a clear description of the effect of the mutation of FANCM on COs in polyploid wheats.

Major comments

The first part of the introduction (L62-85) on recombination process is too much detailed and out of the scope of the paper (mutation of *fancm* on recombination in wheat). On the contrary, nothing is said regarding the distribution of recombination in wheat itself, relationships with sequence features while many papers (including some from the authors but not only, and these latter are missing) have been published. I suggest to completely reconsidering introduction by reducing significantly this part, comment more on wheat recombination and focus more on the effect of mutation of *fancm* on recombination without dealing with the mechanistic. The authors should keep the mechanisms for the discussion to support their conclusions.

In "Materials and Methods" section, a part is missing concerning the bioanalysis of the sequences. How are the sequences from the various genes retrieved? From which initial sequence?

Regarding epigenetic marks, it is not clear from the "Materials and Methods" section, if they have been produced specifically for this work or if they derive from previous analyses. If they have been produced here, was it on meiocytes or on a different tissue? If this was on meiocytes, at what stage? Since chromatin evolves during meiosis with condensation/decondensation steps, this may affect chromatin status and this may change the marks. If the marks derive from a different tissue, what evidence do you have that the marks are the same in this tissue and in meiocytes?

In the results, nothing is said concerning the differences that may exist between the different copies. How different are they between homoeologous copies, between Kronos and Cadenza? Is there any mutation/indel that may suggest that one copy is not functional? How different are they from those of models (*Arabidopsis*, rice) suggesting that they have the same function? A short section could be added.

A 15%/36% reduction of seed set is observed in hexaploid/tetraploid respectively. Did you check the viability of the seeds as well as their chromosome number? Did you make controls by crossing mutants with a wild type (both ways) to see the female fertility and the true male fertility?

I am not convinced by immunolabelling with Mus81 antibody. Resolution of the image is not good enough to see whether the foci locate on the chromosomes or if it is just background and lack of specificity. For figure 3E, it looks as if the stages are not the same for WT and mutant (pachytene vs zygotene). It seems that there is more chromatin in mutants (Cadenza?) and/or that it is less condensed (zygotene?) while it is previously stated (L410-432) that synapsis is normal. Be sure that these images are appropriate or remove.

Concerning the genetic maps, I do not understand the last sentence L501-502. What is the difference between a cM map-length increase and a cumulative increase? It is also well known that recombination is uneven in wheat with more recombination in distal regions. This is not discussed. From Fig.5C, recombination seems increased in distal part of the chromosome but reduced closer to the pericentromere and not changed at centromere/pericentromere. This is also what is observed in Arabidopsis.

Discussion almost relies on what is common with what is already known from models. Discussion should be more focused on what is the originality of the results in wheat compared to other species.

Minor comments

L67-68: "In Arabidopsis and wheat, COs form via the class I or class II pathways". In all eukaryotes with sexual reproduction, Class I and/or Class II COs are observed.

L123-125: It is surprising that the IDs of genes for durum mutant correspond to Traes. Should it be Trdur? Or are these the same as for aestivum?

L129: "...and characterised a Ttmsh5 null mutant (K863)..." This is not clear. As a tetraploid, there should be two copies of TtMsh5. Does this mean that one copy is not functional? If yes, which one? Specify then which copy the mutant is derived from.

L133: Give the Traes for the A and B copies to keep consistency with the D copy (see my comment above).

L141-150: The same primers were used for both Kronos and Cadenza?

L167: The way the VIGS experiment is performed is surprising. I am not very familiar but usually, the virus is propagated in tobacco leaves and is purified with the sap. Virus is inoculated with the sap through gentle friction. Could you provide references for this interesting protocol?

L168: At what stage the plants are inoculated? This is important as you expect to observe the effect of VIGS two weeks after inoculation. Since FANCM acts during meiosis, it should be two weeks before. But this depends on the genotype.

L319-321: "In Ttfancm_1, 0.98 ± 0.09 ($n = 104$) univalents per cell..." Since multivalents are never mentioned, you should have at least two, four, six... univalents/defective cell. Since you have 64% of the cells that are defective, I am doubtful about this value which should be >1 . Can you check and confirm.

L325-326: How do you count the chiasmata? I suspect that you count the number of rings/rods but this is not indicated in "Materials and Methods" section.

L395-399: The two sentences are almost identical and could be compiled into one.

L435-455: There is a clear inconsistency between Fig.1D and the numbers given in this section questioning the quality of the global analysis. "...ring bivalents accounted for only 11% of chromosome

pairs in Ttmsh5..." Based on Fig.1D, it is rather 3%. "...which significantly increased to 32%..." Again, based on Fig.1D, it is ~20%... According to the estimates from this figure (70% univalents, 27% rods, 3% rings, 194 cells) and considering the way chiasmata should have been counted (not given), in Ttmsh5, a mean of 4.7 chiasmata per cell is correct. For msh5/fancm mutant (20% rings, 45% rods, 35% univalents, 86 cells), 12.4 chiasmata seems also correct. So I guess that the % need to be changed.

L514-526: Since the recombination profile is not changed between WT and mutants, correlation with related marks is obviously not changed as well. I am not sure that this part brings a lot to the paper. It would have been more relevant to look if marks are changed between the mutant and the WT.

Reviewer #3:

Remarks to the Author:

In this manuscript, the authors analyse the consequences of FANCM deficiency on meiotic recombination in wheat. FANCM is the first protein with anti-crossover activity identified in plants. S.D. Desjardins and co-workers show that FANCM is essential for at least one crossover to form between homologous chromosomes in wheat, a necessary condition for proper chromosome segregation. This result echoes what has been observed in Lettuce, where the number of bivalents is reduced when FANCM is removed. Desjardins et al. also observed that total crossovers increased in wheat fancm mutant compared to wild type. Overall, I found the manuscript interesting and well-written, although the effect of FANCM on the obligatory CO has lost some of its originality since the publication on lettuce.

I do have some major concerns, however, which relate mainly to immunostaining and modelling. Immunostaining against MUS81 was conducted using a primary antibody (Ab) raised against TaMUS81, which was first described in a previous paper (Desjardins et al., 2020). Unless I am mistaken, the specificity of this antibody has never been validated. Desjardins et al., (2020) showed that the number of MUS81 foci per meiocyte at pachytene "matched expectations" (15% of total COs, as in Arabidopsis) but this is not sufficient to ensure that the TaMUS81 primary Ab marks (all) sites of class II COs. The very uneven distribution of the MUS81 foci in WT Kronos (Fig4E), where 3 bright points occupy the same very small part of the 2D space, does not convince me. At the very least, I would tone down the conclusions taken from this experiment.

Unfortunately, immunolocalization of TtFANCM is not more convincing. As the authors quite honestly acknowledge (lines 470-471), there is cross-reactivity with the Ab raised against ASY1 (or between secondary Abs??). As clearly demonstrated by Fig4F, there is extensive labelling of chromatin in Ttfancm_1 mutant. As a result, I do not see how to use this Ab to track the dynamics and count the number of FANCM foci. For me, all this part (lines 457-488) is subject to caution.

I am a bit puzzled by the conclusions taken from the GLM analysis. First, it is based on only a few intervals while the final model incorporated a lot of predictors. This imbalance between the number of observations and the number of terms could lead to overfitting issues. Secondly, Figure 5B suggests that the response variable is not randomly distributed along the chromosomes (which makes sense, by the way), with intervals showing an increase being, in general, more distally located than intervals showing a decrease in COs. As the genomic landscape of DNA methylation, H3K27me1 and H3K27ac follows the same trends (Tock et al.), using these variables as predictors will necessarily lead to the conclusion that they are associated with CO change. This does not mean they are causal, and the conclusion that "chromatin influencing changes to crossover rate in wheat fancm mutants" is thus over-stated. Could the authors use the crossover rate (cM/Mb) estimated from wheat genetic maps or an centromere-telomere index as predictors? I would be curious to see how these variable rank among the predictors? If they prove to be as good predictors as DNA methylation, H3K27me1 and H3K27ac, then the conclusion should be different. This would show that the propensity to form OC influences changes in crossover rate in wheat fancm mutants, without presuming the underlying causes.

Other comments

Lines 93-95 : « chiasmata (the cytological manifestations of COs) were significantly reduced, although technical limitations may have precluded detection of more than one closely spaced CO using this analysis.”. I found this "additional information" to be misleading or useless, given the technical inability to detect more than one closely spaced CO. I would simplify the message here.

Line 326-327 and 342-343: There is a 18% decrease in chiasmata per cell in Ttfancm mutant (tetraploid wheat) but a 9% decrease in chiasmata per cell in Tafancm mutant (hexaploid wheat). Likewise, there is on average 1 univalent per cell in Ttfancm and only 0.5 univalent per cell in Tafancm. Intuitively, I would have expected to see a stronger impact of FANCM deficiency in plants with a higher chromosome number. Why is my bet wrong? Is there a way to test whether all mutant alleles in hexaploid wheat are truly null alleles? The fact that “VIGS and TILLING mutants reproduced the same phenotype for chiasma formation” can be interpreted in both directions: either VIGS has completely abolished FANCM activity, or a knock-down mutant has been produced by TILLING.

Line 570-571: the authors conclude that their “data is consistent with the molecular function of FANCM as a DNA helicase that unwinds strand invasion D-loop intermediates during zygotene”. However, they did not observe any difference in the number of HEI10 foci per meiocyte between Ttfancm and wild type *T. turgidum* (line 396) at this stage, whereas early HEI10 foci are thought to mark early recombination intermediates at zygotene. The difference in HEI10 foci numbers between WT and fancm wheat mutant does not appear until later, at the pachytene, when the D-loop has matured into a double Holliday junction. How do the authors explain this lag time? If FANCM “is required to release resected DSB ends from invasion of sister chromatids to enable engagement with the homolog (lines 548-549), why is there the same number of HEI10/MSH5 foci between fancm mutant and WT at zygotene?

Reviewer #4:

Remarks to the Author:

The manuscript by Desjardins and colleagues describes the effect of fancm mutations on wheat meiosis. Both TILLING and VIGS approaches were used and confirmed that fancm loss of function leads to a reduction in obligate CO (explaining the loss in fertility), associated with an increase of class 2 COs.

The paper describes a very nice work, and is very well written.

Because I am not an expert in the mechanisms of meiosis, nor a wheat geneticist, I will focus my comments on the epigenetic aspects of the findings.

1. The authors show that DNA methylation, H3K27me1 and H3K27ac are good predictors of CO rate differences between fancm and WT. Could the authors elaborate on what is known on the role of these marks on CO rates for instance in Arabidopsis? Isn't it paradoxical to have two antagonistic marks on the same H3 residue having similar effects on CO rates in fancm?

2. Probably I missed this point but there seem to be 5 intervals in which the recombination rate is negatively affected by the fancm mutations (Table 1), including one drastically affected on chr5B. Is there something special with these intervals in the wheat genome?

3. Lines 504:512: is the recombination rate in the tetraploid given as a control? The effect seems opposite to Table 1.

Minor point: tetraploid/hexaploid could be indicated on Table 1&2 legends, and the tables could be better detailed (mean etc.).

REVIEWER COMMENTS

Reviewer #1 (Remarks to the Author):

This manuscript submitted by Desjardins et al reports on a study of FANCM's functions in maintaining obligate class I crossovers and suppressing class II crossovers in wheat. By analyzing *fancm* mutants in tetraploid and hexaploid wheat, the authors report that FANCM is required for normal fertility as mutants exhibited increased univalents and reduced chiasmata. Similarly, TaFANCM-i mutants by VIGS showed comparable defects. In contrast to a narrow distribution of chiasmata numbers among wild type cells, mutants display a greater range of chiasma numbers observed. By FISH analysis of rDNA on chromosomes 1B and 6B, the overall reduction of chiasmata in mutants was observed. Authors then showed decreased HEI10 foci in mutants during later stages and elevated numbers of MUS81 foci, suggesting that FANCM plays a role in ensuring class I crossovers and inhibiting class II crossovers. The inhibition of class II crossovers by FANCM was also demonstrated by quadruple mutants of *Ttmsh5 Ttfancm*. FANCM first forms numerous axis-associated foci and, by mid-zygotene, FANCM appears as larger foci that overlapped partially with HEI10 foci. In *Ttmsh5* mutants, FANCM foci were found to reduce by about 51%, which may imply that FANCM may be able to locate on sites other than class I COs. In addition, the authors showed that genetic distances were overall increased in hexaploid *fancm* mutants. Last, the authors showed the correlation between DNA methylation and histone marks with crossover rate intervals.

Overall, this study is well conducted and presented. The Introduction is generally an excellent presentation of the background. The experimental analyses are quite clearly presented. I particularly appreciated the analysis of FANCM in tetraploid and hexaploid wheat, as well as by VIGS. I also appreciate the amount of work the authors put into in genetic mapping and correlation with chromatin marks. My comments and suggestions are listed below.

We thank all the reviewers for recognising the importance of our findings and making helpful comments and suggestions to improve the manuscript. Accordingly, we have endeavoured to modify the manuscript to address the issues raised.

1. The FISH results seem to be less informative and didn't turn into a clear conclusion.

The relevance of the FISH analysis is that both the NOR-bearing chromosomes (1B & 6B) display a reduction in chiasmata in the *fancm* mutant, relative to wild type. 1B & 6B also form inherently fewer chiasmata per bivalent than the other chromosomes in wild type. A direct consequence of this is that, while chromosomes 1B and 6B lose the same proportion of chiasmata as all other chromosomes in *fancm* mutants, they are less capable of buffering against this general loss and are therefore more likely to form univalents. Therefore, the clear conclusion is that the NOR chromosomes are already predisposed to form fewer chiasmata and therefore in the *fancm* mutant are more likely to form univalents than other chromosomes. The last sentence of the paragraph states that 'Therefore, a direct consequence of the *fancm* null mutant appears to be a reduced ability of the NOR chromosomes to buffer against a general loss of chiasmata.'

As the pSc119.2-2 probe also marks other chromosomes, is it possible to include an analysis of other chromosomes?

This was our original intention but unfortunately, we had difficulty with reliable penetration of the pSc119.2-2 probe in meiocytes, which prevented consistent identification of additional

chromosomes and comparison across all cells. Instead, pSc119.2-2 was used to validate the identification of chromosomes 1B and 6B, as it reliably gave signals on these chromosomes.

It will be also helpful if the authors can identify and label some chromosomes in Fig S5.

As suggested, chromosomes 1B and 6B are now indicated on Fig S5.

2. Although there is a notable decrease (20%) in HEI10 foci in mutants at pachytene stage, it still seems ambiguous to say that FANCM directly promotes class I COs.

In the *fancm* mutant, in addition to the decrease in HEI10 foci, we observed an increase in univalents, indicating loss of the obligate CO that forms via the class I CO pathway. Taken together, we feel the data supports the statement that FANCM is needed to promote class I COs in WT.

For example, FANCM may be involved in dissociating the undesired invasions (eg. between sisters or other types of intermediates), so that fewer inter-homolog invasions can be labelled by HEI10?

We acknowledge this possibility, but we detected no change in installation of the synaptonemal complex, indicating that stable strand-invasion was normal. A homeostatic mechanism may be in place obscuring this, so we had suggested that crossing *fancm* with *spo11* hypomorphs as a way to address this question, but this is beyond the scope of the current study. The reasoning would be that the homeostatic control would be perturbed with fewer DSBs.

In addition, please provide reference to support the implication of “localization of HEI10 may represent double Holliday junctions” (lines 407-408).

Following the above reviewer’s comment we have decided to moderate our comment in the text. As HEI10 is a marker of Class I COs during late pachytene/diakinesis we have restricted ourselves to stating that the reduction of HEI10 foci suggests that some Class I COs are sensitive to the loss of FANCM (see below). Evidence from mouse and yeast both indicate that HEI10/Zip3 interacts with double Holliday junctions (Refs: *Qiao H, Rao HBDP, Yang Y, Fong JH, Cloutier JM, Deacon DC, Nagel KE, Swartz RK, Strong ER, Holloway JK, et al. 2014. Antagonistic roles of ubiquitin ligase HEI10 and SUMO ligase RNF212 regulate meiotic recombination. Nat Genet 46: 194–199; Lynn et al 2007*)

However while it is likely that the protein will behave similarly in plants formal evidence of has not been obtained thus far. Hence we have now included the following line: “This reduction in HEI10 foci at pachytene suggests that formation of a proportion of Class I COs is sensitive to the loss of FANCM in wheat”.

3. As the authors acknowledge, the chiasma numbers may be underestimated. I am still concerned that the numbers of chiasmata are much lower than observed HEI10 foci (only for CO I). Thus, I wonder how reliable the chiasma analyses are.

The chiasma analysis is an established and routinely used technique to measure the minimum number of COs which is useful for comparing two different genotypes. Numerous studies have shown that chiasmata directly correspond to CO sites (the classic example being the study by Charles Tease C. *Nature* 1978 272(5656):823-4. doi: 10.1038/272823a0). While scoring well-spaced chiasmata (say one in each chromosome arm) is straightforward, it is difficult to detect multiple chiasmata in a single chromosome arm. Hence a conservative strategy is routinely adopted whereby a rod bivalent is scored as single chiasma/CO and a ring bivalent scored as two chiasmata/COs.

Additional chiasmata are only included if based on bivalent shape they are obvious. This gives a minimum number of chiasmata but does likely underestimate the total number of CO events. The approach we used is described and based on Osman et al 2021.

A further consideration is that the numbers of HEI10 foci are dynamic and begin with a large number that decline to a smaller number that mark COs at late pachytene (Osman et al 2021). Studies in *Sordaria* (de Muyt G&D 2014 vol28 p1111) have shown that the numbers of HEI10 foci are dynamic during pachytene with the more numerous T2 foci being replaced at mid-pachytene by the larger (Class I CO specific) T3 foci. Our studies in *Arabidopsis* (Lambing et al, 2014) also show that the size difference between the classes of HEI10 foci is not that great (T2=175nm - T3=250+nm). Hence, any slight variation in the stage of pachytene between cells could lead to an elevation in the estimation of HEI10 foci present. This is compounded in wheat since as reported by Osman et al 2021, the temporal-spatial progression of meiotic prophase I within each meiocyte is skewed with more distal regions progressing to later stages of prophase I before interstitial/distal regions. Together these factors may result in a slight tendency to bias HEI10 numbers upwards.

4. Since one HEI10 antibody produces a linear signal on unsynapsed chromosomes, is it possible that the FANCM signals on unsynapsed signals are real?

The focal signal is the true FANCM signal, while the unsynapsed linear FANCM is an artefact brought about by cross-reactivity with anti-ASY1. There are two main reasons that we conclude this. 1) The anti-FANCM antibody only produces the secondary linear signal on unsynapsed chromosomes when used in combination with anti-ASY1 and not other antibodies. When anti-FANCM is used exclusively, or in combination with other antibodies, such as anti-HvHE10 and/or anti-AtZYP1, the linear signal disappears, while the primary focal signal persists. 2) In the *Ttfancm_1* null mutant the FANCM focal signal disappears, while the linear signal persists (again only when used in combination with anti-TaASY1).

Another related question is whether the FANCM antibody can still recognize truncated proteins in *Ttfancm_1* mutant? (Fig 4F)

The TILLING mutations (Kronos 3873; Kronos 0234) result in premature STOP codons and the potential for truncated proteins in the *Ttfancm_1* mutant. However, these putative truncated proteins are upstream of antigen used to generate the anti-FANCM polyclonal antibody, so the anti-FANCM antibody should not recognise the truncated proteins, if they are present.

Reviewer #2 (Remarks to the Author):

Importance of the research

Meiotic recombination events (crossovers) are at the heart of genetic improvement as they ensure a faithful production of fertile gametes and they allow shuffling of alleles to create new innovative and powerful combinations. However, crossovers are rare events (at least one mandatory but rarely more than three per homologous pair) and there is therefore an interest to improve the rate in crops to speed up the development of performing varieties. Several genes that increase recombination rate have been found in the model species *Arabidopsis* and it is crucial to estimate how these genes affect recombination in crops, especially in polyploids like wheat. Among these genes is the helicase FANCM (Fanconi anemia complementation group M) that shows contradictory results in

Arabidopsis: *fanm* mutation increases recombination in inbred lines but has no effect in hybrid context. Interestingly, *fanm* mutation increases significantly recombination in Brassica, rice and pea. It is thus of main interest to see whether similar results are observed in wheat.

We thank the reviewer for placing our work in context.

Noteworthy results

Impact of the mutation of *fanm* in tetraploid and hexaploid wheats is detailed. Fertility of the double and triple mutants is reduced compared to wild types and simultaneous reduced viability of pollen suggests a default during gamete formation. Double mutation of *fanm* in a tetraploid background reduces the number of chiasmata. This reduction resulted from the presence of univalents at metaphase I leading to further chromosome mis-segregation. Same result was observed in hexaploid *fanm* mutants and this was confirmed using a VIGS approach. Use of HEI10 antibody shows that type I COs are slightly reduced in *fanm* mutants at late prophase I only suggesting that at this late stage, location of HEI10 on chromosomes depends on FANCM. The number of double strand breaks as well as synapsis and meiotic progression were not affected in mutants. Implication of FANCM in class II COs was confirmed using double *msh5/fanm* mutants and FANCM was immunolocalized on chromosomes using a wheat-specific antibody. Mutation of *fanm* resulted in a significant increase of genetic distances in hexaploid context but was not changed in the tetraploid. Overall, this study provides a clear description of the effect of the mutation of FANCM on COs in polyploid wheats.

Major comments

The first part of the introduction (L62-85) on recombination process is too much detailed and out of the scope of the paper (mutation of *fanm* on recombination in wheat).

We have removed the section (below). We wrote this section because our *fanm* results will be of interest to the general field working on recombination. Relating this work to the mechanism in yeast may explain some of the results, but this may be more suited in the discussion.

These apparently contradictory phenotypes may be explained by an analysis performed in budding yeast with the FANCM ortholog, *Mph1*. *Mph1* promotes CO formation by dissociating the resected DSB from the sister-chromatid, thus allowing it to invade the homologous chromosome²⁸. In *mph1* mutants, inter-sister repair events were elevated, but the number of COs remained the same indicating that a homeostatic process preferentially repaired inter-homolog interactions as COs over NCOs²⁸. Homeostatic rebound may overcompensate beyond wild-type levels, thus resulting in additional COs in certain organisms²⁸, but fewer inter-homologue interactions may limit accurate chromosome pairing and CO formation.

On the contrary, nothing is said regarding the distribution of recombination in wheat itself, relationships with sequence features while many papers (including some from the authors but not only, and these latter are missing) have been published. I suggest to completely reconsidering introduction by reducing significantly this part, comment more on wheat recombination and focus more on the effect of mutation of *fanm* on recombination without dealing with the mechanistic. The authors should keep the mechanisms for the discussion to support their conclusions.

We acknowledge that we could provide further description of the crossover landscape in wheat and known genomic and epigenomic correlates, so we have added the line:

COs correlate with enrichment of the Polycomb histone modification H3K27me3 at the distal chromosome ends, indicating a potential role for facultative heterochromatin in shaping the recombination landscape (Tock et al. 2021).

In “Materials and Methods” section, a part is missing concerning the bioanalysis of the sequences. How are the sequences from the various genes retrieved? From which initial sequence?

Thank you for this suggestion. We have added the following section to the methods:

Identification of wheat FANCM

Wheat *FANCM* orthologues were identified using the *Arabidopsis thaliana* amino acid sequences (At1g35530) to BLAST against publicly available databases: *Triticum turgidum* (Maccaferri et al., 2019) Svevo.v1 https://plants.ensembl.org/Triticum_turgidum) and *T. aestivum* (Appels et al., 2018) IWGSC https://plants.ensembl.org/Triticum_aestivum). Wheat cds were aligned using the Clustal W algorithm (gap open cost = 12, gap extend cost = 3), translated and the primary sequence ran through the Conserved Domain Database (NCBI).”

Regarding epigenetic marks, it is not clear from the “Materials and Methods” section, if they have been produced specifically for this work or if they derive from previous analyses. If they have been produced here, was it on meiocytes or on a different tissue? If this was on meiocytes, at what stage? Since chromatin evolves during meiosis with condensation/decondensation steps, this may affect chromatin status and this may change the marks. If the marks derive from a different tissue, what evidence do you have that the marks are the same in this tissue and in meiocytes?

We apologise for this omission. For this analysis, we used published datasets derived from Chinese Spring (ASY1, DMC1, H3K4me1, H3K4me3, H3K27ac, H3K27me3, H3K36me3, H3K9me2, H3K27me1, CENH3 ChIP-seq and BS-seq). Histone ChIP-seq and BS-seq data are derived from seedlings or leaf tissue (Guo et al. 2016 PLoS Genet; IWGSC 2018 Science; Li et al. 2019 Genome Biol; Tock et al. 2021 Genome Res), and ASY1 and DMC1 ChIP-seq data are derived from immature pre-emergence spikes (Tock et al. 2021 Genome Res). We fully acknowledge the reviewer’s point that ideally these datasets would be generated from meiotic cells. However, at this time these datasets do not exist.

However, we also note that extensive analysis of meiotic recombination in *Arabidopsis*, and comparison to somatic chromatin datasets, reveals strong correlations. Therefore, although we acknowledge that this is not the perfect comparison, we still expect meaningful relationships to be revealed.

In the results, nothing is said concerning the differences that may exist between the different copies. How different are they between homoeologous copies, between Kronos and Cadenza? Is there any mutation/indel that may suggest that one copy is not functional? How different are they from those of models (*Arabidopsis*, rice) suggesting that they have the same function? A short section could be added.

We have added a section to the results to address this concern:

FANCM is conserved in wheat

To identify *FANCM* orthologues in wheat, BLAST searches were performed using the *Arabidopsis thaliana* *FANCM* amino acid sequence. Two *FANCM* orthologues were identified in tetraploid wheat *T. turgidum*: *TtFANCM-A1* (*TRITD4Av1G171480*) and *TtFANCM-B1* (*TRITD4Bv1G035000*). Three *FANCM* orthologues were identified in hexaploid wheat *T. aestivum*: *TaFANCM-A1* (*TraesCS4A02G217700*), *TaFANCM-B1* (*TraesCS4B02G096400*) and *TaFANCM-D1* (*TraesCS4D02G092800*). All copies are full-length and predicted to produce functional proteins. *FANCM* is highly conserved between ploidy levels with *FANCM-A1* and *FANCM-B1* primary amino acid sequences being identical in tetraploid and hexaploid wheat (1447/1447 & 1458/1458, respectively). While the three *FANCM* homeologues, *FANCM-A1*, *FANCM-B1* and *FANCM-D1*, share 94.2% amino acid identity, with polymorphisms at 85 residues (1377/1462). A consensus sequence was created from the three wheat homeologues to compare with *Arabidopsis*. Wheat *FANCM* shares 38.8% (543/1502) overall amino acid identity with *Arabidopsis*, with increased homology in the predicted DEXDc (DEAD-like helicases superfamily; 70.7%; 135/191) and HELICc (helicase superfamily c-terminal; 67.9%; 93/137) domains.”

A 15%/36% reduction of seed set is observed in hexaploid/tetraploid respectively. Did you check the viability of the seeds as well as their chromosome number?

Yes we did. The tetraploid *Ttfancm* mutant was maintained as both homozygote (aabb) and segregating heterozygote (Aabb & aabB) lines, while the hexaploid *Tafancm* mutant was maintained solely as a homozygote (aabbdd). Seed from tetraploid and hexaploid *fancm* null mutants were viable with good germination efficiency and all observed progeny were euploid.

Did you make controls by crossing mutants with a wild type (both ways) to see the female fertility and the true male fertility?

This is a good suggestion, but we haven't performed an experiment to determine the effect on female fertility.

I am not convinced by immunolabelling with Mus81 antibody. Resolution of the image is not good enough to see whether the foci locate on the chromosomes or if it is just background and lack of specificity. For figure 3E, it looks as if the stages are not the same for WT and mutant (pachytene vs zygotene). It seems that there is more chromatin in mutants (Cadenza?) and/or that it is less condensed (zygotene?) while it is previously stated (L410-432) that synapsis is normal. Be sure that these images are appropriate or remove.

We agree that the specificity of the anti-MUS81 antibody, previously reported in Desjardins et al., (2020), is uncertain, as we do not have a *Tamus81* null mutant on which to test it. We are removing the MUS81 immunolocalisation data as advised. This does not affect the conclusions drawn by the study as the *fancm msh5* double mutant, which is a more reliable indicator of the increase in the class II pathway.

Concerning the genetic maps, I do not understand the last sentence L501-502. What is the difference between a cM map-length increase and a cumulative increase?

The 'cM map length increase' of 19.1% is calculated as the percentage increase of the total *fancm* cM distance over WT control across all regions [from Table 1: $\Sigma_{\text{control}} = 544.1$ cM and $\Sigma_{\text{fancm}} = 647.91$ cM ; $(647.91-544.1)/ 544.1 = 19.1\%$], whereas the 'cumulative percentage increase' describes the

average of the percentage difference of all chromosome regions. We have added the sentence (below) to the online methods:

The '*cM map length increase*' was calculated as the percentage increase of the total *fancm* cM distance over WT control across all regions and the '*cumulative percentage increase*' describes the average of the percentage difference of all chromosome regions.

We have revised the wording to clarify this.

The map length was increased by 19.1% from *fancm* over WT when adding the cM distances across all regions, or by 31.3% when averaging the percentage cM increase from the 21 intervals.

It is also well known that recombination is uneven in wheat with more recombination in distal regions. This is not discussed. From Fig.5C, recombination seems increased in distal part of the chromosome but reduced closer to the pericentromere and not changed at centromere/pericentromere. This is also what is observed in Arabidopsis.

We agree that there are similarities with the change reported in Arabidopsis. We have added the sentence (below) to the discussion.

The distal regions experienced the greatest increase in COs whereas the interstitial/proximal regions were more likely to be lower, consistent with recombination increases in the barley *recq4* mutant (Arrieta et al 2021).

Discussion almost relies on what is common with what is already known from models. Discussion should be more focused on what is the originality of the results in wheat compared to other species.

We have further developed our Discussion section to emphasize the differences in genetic control of recombination that we have uncovered in wheat, compared to other species.

Minor comments

L67-68: "In Arabidopsis and wheat, COs form via the class I or class II pathways". In all eukaryotes with sexual reproduction, Class I and/or Class II COs are observed.

Assuming we understand correctly, we believe this referee's comment is incorrect. Although it is true that most sexually reproducing eukaryotes form class I and class II COs, there are important exceptions. For example in *S. pombe* COs do not form via the class I CO pathway and are interference insensitive. *C. elegans* and *N. crassa* show robust interference and form via COs via the class I pathway. We therefore cannot assume that all eukaryotes possess the class I and class II CO pathways.

L123-125: It is surprising that the IDs of genes for durum mutant correspond to Traes. Should it be Trdur? Or are these the same as for aestivum?

The original reason for this discrepancy was that the Kronos TILLING database is correlated to the C6 hexaploid genome sequence, but we thank you for this suggestion and agree that a change to tetraploid gene models avoids any possible confusion. The IDs of genes for durum mutant now correspond to TRITD: TRITD4Av1G171480 and TtFANCM-B1 TRITD4Bv1G035000.

L129: "...and characterised a Ttmsh5 null mutant (K863)..." This is not clear. As a tetraploid, there

should be two copies of TtMsh5. Does this mean that one copy is not functional? If yes, which one? Specify then which copy the mutant is derived from.

TtMSH5B is a pseudogene and has a natural loss-of-function deletion, which means that mutation of TtMSH5A alone results in a null phenotype (Desjardins et al 2020). The methods have been amended to clarify this:

“We also generated double null mutants: *Ttfancm_1* (K3873 x K3842) and *Ttfancm_2* (K0234 x K3842) and characterised a *Ttmsh5-A1* mutant (K863), which is functionally a null (*Ttmsh5*) as there is a natural loss-of-function deletion in *TtMSH5-B1* (Desjardins et al. 2020), and a *Ttfancm Ttmsh5* double mutant (K3873 x K3842 x K863).”

L133: Give the Traes for the A and B copies to keep consistency with the D copy (see my comment above).

TraesCS4A02G217700 and *TraesCS4B02G096400* have been added.

L141-150: The same primers were used for both Kronos and Cadenza?

Yes, the same primers were used for Kronos and Cadenza. Ta has been removed from the primer names and the text updated for clarification:

“The coding sequences for both varieties were amplified and sequenced using homoeolog-specific primers: FANCM_A_F 5'-CTGGATGTTGGCTGCACTCG-3' and FANCM_A_R 5'-ATGTGGTTGCTTTTCAGAGGTA-3', FANCM_B_F 5'-AGAGGCTATGTTTCTATCACCC-3' and FANCM_B_R 5'-GATCCTGATGTATTCCCTACC-3', FANCM_D_F 5'-AAGGAAGGAAGTGGGAAAGT-3' and FANCM_D_R 5'-CCAGGCAAGCATGAATATCC-3'.”

L167: The way the VIGS experiment is performed is surprising. I am not very familiar but usually, the virus is propagated in tobacco leaves and is purified with the sap. Virus is inoculated with the sap through gentle friction. Could you provide references for this interesting protocol?

We also used the standard procedure as the reviewer describes, but optimised for wheat meiosis following the protocol of Desjardins et al (2020). The information pertaining to accumulation of the virus in the intermediate host has now been added to the online methods.

“The recombinant pCa- γ bLIC, as well pCaBS- α and pCaBS- β , were then transformed into electrocompetent *Agrobacterium tumefaciens* (GV3101), agroinfiltrated and the virus accumulated in intermediate host *Nicotiana benthamiana*, prior to inoculation of wheat plants with infected sap at the 4-4.5 leaf stage (~28 days post-sowing)”

L168: At what stage the plants are inoculated? This is important as you expect to observe the effect of VIGS two weeks after inoculation. Since FANCM acts during meiosis, it should be two weeks before. But this depends on the genotype.

The plants were inoculated two weeks prior to meiosis, which corresponded to the 4-4.5 leaf stage in 'Bobwhite' (~28 days post-sowing). A time course experiment in the category 3 growth facility was conducted by Desjardins et al. (2020). We have added this information to the MS.

“The recombinant pCa- γ bLIC, as well pCaBS- α and pCaBS- β , were then transformed into electrocompetent *Agrobacterium tumefaciens* (GV3101), agroinfiltrated and accumulated in

intermediate host *Nicotiana benthamiana*, and infected sap used to inoculate wheat plants at the 4-4.5 leaf stage (~28 days post-sowing)”

L319-321: “In *Ttfanm_1*, 0.98 ± 0.09 ($n = 104$) univalents per cell...” Since multivalents are never mentioned, you should have at least two, four, six... univalents/defective cell. Since you have 64% of the cells that are defective, I am doubtful about this value which should be >1 . Can you check and confirm.

The reviewer is correct, the 0.98 ± 0.09 ($n = 104$) should refer to *pairs of* univalents per cell. The MS has been updated accordingly:

“In *Ttfanm_1*, 0.98 ± 0.09 ($n = 104$) pairs of univalents per cell were observed at MI, compared to only 0.05 ± 0.03 ($n = 60$) in wild type ($p < 0.001$, Pairwise Wilcoxon Rank Sum Test), and 64% of cells possessed at least one univalent pair in *Ttfanm_1*, compared with only 5% in wild type.”

L325-326: How do you count the chiasmata? I suspect that you count the number of rings/rods but this is not indicated in “Materials and Methods” section.

This has now been added to the methods.

“Chiasmata number was interpreted according to bivalent shape after Osman et al. (2021). Rod bivalents were scored as a minimum of 1 and ring bivalents scored as a minimum of 2, with additional chiasmata scored based on shape.”

L395-399: The two sentences are almost identical and could be compiled into one.

We have merged the two sentences into:

Furthermore, at diakinesis HEI10 foci reduced by 29%, from 34 ± 0.5 ($n = 54$) in wild type to 24 ± 0.5 ($n = 60$) in *Ttfanm_1* (Mann-Whitney U Test, $p < 0.001$) (Fig. 2) and this also occurred in hexaploid wheat.

L435-455: There is a clear inconsistency between Fig.1D and the numbers given in this section questioning the quality of the global analysis. “...ring bivalents accounted for only 11% of chromosome pairs in *Ttmsh5*...” Based on Fig.1D, it is rather 3%. “...which significantly increased to 32%...” Again, based on Fig.1D, it is ~20%... According to the estimates from this figure (70% univalents, 27% rods, 3% rings, 194 cells) and considering the way chiasmata should have been counted (not given), in *Ttmsh5*, a mean of 4.7 chiasmata per cell is correct. For *msh5/fanm* mutant (20% rings, 45% rods, 35% univalents, 86 cells), 12.4 chiasmata seems also correct. So I guess that the % need to be changed.

Thank you, the reviewer is correct, the figures 11% & 32% are errors. These refer to the % of bivalents formed that are rings, not the total number of chromosome pairs (including univalent pairs) that are rings. We have amended this in the manuscript and it now reads:

“Furthermore, ring bivalents accounted for only **3.2%** of chromosome pairs in *Ttmsh5*, which significantly increased to **21%** in *Ttmsh5 Ttfanm* ($p < 0.001$, Pairwise Wilcoxon Rank Sum Test).”

L514-526: Since the recombination profile is not changed between WT and mutants, correlation with related marks is obviously not changed as well. I am not sure that this part brings a lot to the paper.

We have included a new correlation analysis that compares the differential crossover rate in *fancm* relative to wild type with euchromatic, heterochromatic and recombination markers computed within each genetic marker interval. We observe that differential crossover rate shows a significant positive correlation with a previously published wild type crossover rate map derived from a Chinese Spring x Renan recombinant inbred population (IWGSC 2018 Science). Differential crossover rate shows non-significant positive relationships with ChIP-seq signals for the chromosome axis protein ASY1 and the meiotic recombinase DMC1, the euchromatic marks H3K4me1, H3K4me3 and H3K27ac, and the Polycomb mark H3K27me3, and non-significant negative correlations with H3K9me2, H3K27me1, centromeric CENH3, and DNA methylation in CG and CHG contexts. From this we conclude that crossovers are increased in *fancm*, but they appear largely where the wild type crossovers are. This implies that chromatin shapes the recombination landscape in similar ways in both wild type and *fancm*, which we think is worth reporting. We have included a new heatmap showing these relationships in SFig. 10.

SFigure 10. Spearman's rank-order correlation coefficients (r_s) for the indicated parameter pairs computed within each genetic marker interval. Correlation coefficients are indicated by cell colour. P -values for r_s correlation coefficients were standardized to represent those based on pairwise values across 100 marker intervals and are indicated within each cell. Included data sets are differential crossover rate (*fancm* cM/Mb - wild type cM/Mb; "Diff_cMMb"), wild type crossover rate derived from a Chinese Spring x Renan genetic map ("IWGSC_cMMb") (IWGSC 2018), ASY1, DMC1, H3K4me3, H3K9me2 and H3K27me1 ChIP-seq (Tock et al. 2021), H3K4me1 and H3K27ac ChIP-seq (Li et al. 2019), H3K27me3 and H3K36me3 ChIP-seq (IWGSC 2018), CENH3 ChIP-seq (Guo et al. 2016), whole-genome bisulfite sequencing-derived DNA methylation (mCG, mCHH and mCHG proportions) (IWGSC 2018), and the distance between the midpoint of each marker interval and the midpoint of previously defined centromeric coordinates ("Dist_to_CEN") (IWGSC 2018).

It would have been more relevant to look if marks are changed between the mutant and the WT. This is an interesting question. Our assumption is that the recombination pathway is downstream of chromatin. However, we acknowledge the reviewer's point that chromatin may also change in recombination mutants. To fully address this point it would be necessary to perform chromatin profiling specifically during meiosis, which is beyond the scope of this study.

Reviewer #3 (Remarks to the Author):

In this manuscript, the authors analyse the consequences of FANCM deficiency on meiotic recombination in wheat. FANCM is the first protein with anti-crossover activity identified in plants. S.D. Desjardins and co-workers show that FANCM is essential for at least one crossover to form between homologous chromosomes in wheat, a necessary condition for proper chromosome segregation. This result echoes what has been observed in Lettuce, where the number of bivalents is reduced when FANCM is removed. Desjardins et al. also observed that total crossovers increased in wheat *fancm* mutant compared to wild type. Overall, I found the manuscript interesting and well-written, although the effect of FANCM on the obligatory CO has lost some of its originality since the publication on lettuce.

We thank the reviewer for noting the value of our work. We would also like to point out that our study goes beyond the referenced studies.

I do have some major concerns, however, which relate mainly to immunostaining and modelling. Immunostaining against MUS81 was conducted using a primary antibody (Ab) raised against TaMUS81, which was first described in a previous paper (Desjardins et al., 2020). Unless I am mistaken, the specificity of this antibody has never been validated. Desjardins et al., (2020) showed that the number of MUS81 foci per meiocyte at pachytene "matched expectations" (15% of total COs, as in Arabidopsis) but this is not sufficient to ensure that the TaMUS81 primary Ab marks (all) sites of class II COs. The very uneven distribution of the MUS81 foci in WT Kronos (Fig4E), where 3 bright points occupy the same very small part of the 2D space, does not convince me. At the very least, I would tone down the conclusions taken from this experiment.

Reviewer #2 shared this concern. We agree that the specificity of the anti-MUS81 antibody, previously reported in Desjardins et al., (2020), is uncertain as we do not have a *Tamus81* null mutant on which to test it. We are therefore removing the MUS81 immunolocalisation as advised. This does not affect the conclusions drawn by the study as the *fancm msh5* double mutant is a more reliable indicator of the increase in the class II pathway.

Unfortunately, immunolocalization of TtFANCM is not more convincing. As the authors quite honestly acknowledge (lines 470-471), there is cross-reactivity with the Ab raised against ASY1 (or between secondary Abs??). As clearly demonstrated by Fig4F, there is extensive labelling of chromatin in Ttfan_{cm}_1 mutant. As a result, I do not see how to use this Ab to track the dynamics and count the number of FANCM foci. For me, all this part (lines 457-488) is subject to caution.

In the current study we present a novel anti-TaFANCM antibody. This antibody localises to axis-associated foci, most notably throughout zygotene. The reviewer has expressed concern that when used in combination with anti-TaASY1 (used to stage the meiocytes) a secondary linear signal is observed, most likely from cross-reactivity with ASY1, which may hinder observation of FANCM foci throughout prophase I. We concede that this background artefact is unfortunate. However, we are confident that the focal signal generated by this antibody is a fair representation of FANCM localization dynamics. Our confidence in this signal is fivefold. 1) The anti-FANCM antibody only produces the secondary linear signal on unsynapsed chromosomes when used in combination with anti-TaASY1. In other experiments where anti-FANCM is used exclusively or in combination with other antibodies, such as anti-HvHE10 and/or anti-AtZYP1, the linear signal is not present, while the primary focal signal persists. A clear representation of the anti-FANCM antibody in the absence of anti-ASY1 can be seen in Fig4H (in red), as axis-associated foci at zygotene. 2) In pachytene nuclei where the entire chromosomes are synapsed and ASY1 has been dissociated, no linear signal is observed and only the focal signal persists. 3) In the Ttfan_{cm}_1 null mutant the primary FANCM focal signal disappears (as the antigen is downstream of the premature STOP codons), while the secondary linear signal persists (again only when used in combination with anti-TaASY1). 4) The FANCM foci observed with this antibody are clean, axis-associated, consistent and reproducible. 5) The number of FANCM foci was significantly reduced in *msh5* null mutants.

This submission does not rely on the FANCM antibody for its main conclusions, but we believe the antibody has been tested thoroughly and the conclusion drawn from its use relevant and of interest to the scientific community.

The anti-FANCM antibody only produces the secondary linear signal on unsynapsed chromosomes when used in combination with anti-ASY1. In other experiments where anti-FANCM is used exclusively or in combination with other antibodies, such as anti-HvHE10 and/or anti-AtZYP1, the linear signal disappears, while the primary focal signal persists. 2) In the Ttfan_{cm}_1 null mutant the true FANCM focal signal disappears, while the secondary linear signal persists (again when used in combination with anti-TaASY1).

We concede that this secondary tracking of the unsynapsed axes by the FANCM antibody is unfortunate, but despite this we have confidence in the focal signal, which is clearly distinct, readily distinguishable, convincing and consistent.

The TILLING mutations (Kronos 3873; Kronos 0234) that result in premature STOP codons and truncated proteins in the Ttfan_{cm}_1 mutant, are upstream of antigen used to generate the anti-FANCM polyclonal antibody, so the anti-FANCM antibody cannot recognise the truncated proteins.

I am a bit puzzled by the conclusions taken from the GLM analysis. First, it is based on only a few intervals while the final model incorporated a lot of predictors. This imbalance between the number of observations and the number of terms could lead to overfitting issues. Secondly, Figure 5B suggests that the response variable is not randomly distributed along the chromosomes (which

makes sense, by the way), with intervals showing an increase being, in general, more distally located than intervals showing a decrease in COs. As the genomic landscape of DNA methylation, H3K27me1 and H3K27ac follows the same trends (Tock et al.), using these variables as predictors will necessarily lead to the conclusion that they are associated with CO change. This does not mean they are causal, and the conclusion that “chromatin influencing changes to crossover rate in wheat *fancm* mutants” is thus over-stated. Could the authors use the crossover rate (cM/Mb) estimated from wheat genetic maps or an centromere-telomere index as predictors? I would be curious to see how these variable rank among the predictors? If they prove to be as good predictors as DNA methylation, H3K27me1 and H3K27ac, then the conclusion should be different. This would show that the propensity to form OC influences changes in crossover rate in wheat *fancm* mutants, without presuming the underlying causes.

Thank you for raising these concerns about possible overfitting given the relatively small number of marker intervals. To address this we selected a smaller number of predictor variables by removing those that show multi-collinearity, which occurs when predictor variables are highly correlated with each other. ASY1, H3K4me3, H3K36me3, H3K9me2, CHG-context DNA methylation, and several interaction terms were removed from the model due to their high variance-inflation factors (VIFs), indicative of multi-collinearity. In addition, we applied regularized regression using the elastic net method from the caret package in R, which penalizes large coefficients in order to avoid overfitting. We then evaluated model performance on training and test subsets of the genetic marker intervals. Training sets were defined as randomly selected subsets of the data, constituting 70% of the intervals, and test sets were defined as the remaining 30% of the intervals. However, the model did not generalize well to the test sets, as indicated by higher root mean squared error (RMSE) and lower R-squared values. Similarly, the model did not generalize well to test sets that were defined as marker intervals within each chromosome, also indicating overfitting. Consequently, we have removed the model from the manuscript.

As above, we include a new correlation analysis that compares the differential crossover rate in *fancm* relative to wild type with euchromatic, heterochromatic and recombination markers computed within each genetic marker interval. We observe that genomic regional variation in crossover rate differences between *fancm* and wild type shows a significant positive correlation with a previously published wild type crossover rate map derived from a Chinese Spring x Renan recombinant inbred population (IWGSC 2018 Science). Differential crossover rate in *fancm* also shows non-significant positive relationships with ChIP-seq signals for the chromosome axis protein ASY1 and the meiotic recombinase DMC1, the euchromatic marks H3K4me1, H3K4me3 and H3K27ac, and the Polycomb mark H3K27me3, and non-significant negative correlations with H3K9me2, H3K27me1, centromeric CENH3, and DNA methylation in CG and CHG contexts. Therefore, while crossovers are increased in *fancm*, their genomic distribution is comparable to that in wild type. This suggests that chromatin may influence the recombination landscape in similar ways in both wild type and *fancm*. We have included a new heatmap showing these relationships in SFig. 10.

Other comments

Lines 93-95 : « chiasmata (the cytological manifestations of COs) were significantly reduced, although technical limitations may have precluded detection of more than one closely spaced CO using this analysis.”. I found this "additional information" to be misleading or useless, given the technical inability to detect more than one closely spaced CO. I would simplify the message here. We feel this statement is useful for the reader regarding the chiasma analysis in the results.

Line 326-327 and 342-343: There is a 18% decrease in chiasmata per cell in Ttfan^{cm} mutant (tetraploid wheat) but a 9% decrease in chiasmata per cell in Tafan^{cm} mutant (hexaploid wheat). Likewise, there is on average 1 univalent per cell in Ttfan^{cm} and only 0.5 univalent per cell in Tafan^{cm}. Intuitively, I would have expected to see a stronger impact of FANCM deficiency in plants with a higher chromosome number. Why is my bet wrong?

We don't know the answer to this but can speculate that gene dosage may play a part. For example, in tetraploid wheat MSH5A is functional whereas MSH5B has been naturally mutated. In the hexaploid, MSH5D is also functional, but MSH4D is mutated. So a greater dosage of MSH5 in the hexaploid may stabilise class I COs, compared to tetraploid wheat.

Is there a way to test whether all mutant alleles in hexaploid wheat are truly null alleles? The fact that "VIGS and TILLING mutants reproduced the same phenotype for chiasma formation" can be interpreted in both directions: either VIGS has completely abolished FANCM activity, or a knock-down mutant has been produced by TILLING.

The TILLING mutations produce Stop codons that consequently truncate the FANCM proteins. Our FANCM antibody is not raised against the truncated part, so we would not detect truncated proteins if they were to localise to chromosomes. However, there is no literature suggesting that truncated FANCM proteins are functional and Crismani et al (2012) also used a similar EMS approach to create null mutants of fan^{cm}.

Line 570-571: the authors conclude that their "data is consistent with the molecular function of FANCM as a DNA helicase that unwinds strand invasion D-loop intermediates during zygotene". However, they did not observe any difference in the number of HEI10 foci per meiocyte between Ttfan^{cm} and wild type *T. turgidum* (line 396) at this stage, whereas early HEI10 foci are thought to mark early recombination intermediates at zygotene. The difference in HEI10 foci numbers between WT and fan^{cm} wheat mutant does not appear until later, at the pachytene, when the D-loop has matured into a double Holliday junction. How do the authors explain this lag time? If FANCM "is required to release resected DSB ends from invasion of sister chromatids to enable engagement with the homolog (lines 548-549), why is there the same number of HEI10/MSH5 foci between fan^{cm} mutant and WT at zygotene?

We hypothesise that the defect occurs during zygotene so that the effect on HEI10 is observed at pachytene. The defect may also be at pachytene, but if plants are similar to yeast, we would expect double Holliday Junctions to have formed (Lynn et al 2007) and therefore there would be no need for FANCM at that point.

Reviewer #4 (Remarks to the Author):

The manuscript by Desjardins and colleagues describes the effect of fan^{cm} mutations on wheat meiosis. Both TILLING and VIGS approaches were used and confirmed that fan^{cm} loss of function leads to a reduction in obligate CO (explaining the loss in fertility), associated with an increase of class 2 COs.

The paper describes a very nice work, and is very well written.

Because I am not an expert in the mechanisms of meiosis, nor a wheat geneticist, I will focus my comments on the epigenetic aspects of the findings.

1. The authors show that DNA methylation, H3K27me1 and H3K27ac are good predictors of CO rate differences between *fancm* and WT. Could the authors elaborate on what is known on the role of these marks on CO rates for instance in Arabidopsis? Isn't it paradoxical to have two antagonistic marks on the same H3 residue having similar effects on CO rates in *fancm*?

Due to overfitting, we have removed this model from the manuscript and replace it with a correlation analysis that compares the differential crossover rate in *fancm* relative to wild type with euchromatic, heterochromatic and recombination markers computed within each genetic marker interval. We observe that genomic regional variation in crossover rate differences between *fancm* and wild type shows a significant positive correlation with a previously published wild type crossover rate map derived from a Chinese Spring x Renan recombinant inbred population (IWGSC 2018 Science). Differential crossover rate in *fancm* also shows non-significant positive relationships with ChIP-seq signals for the chromosome axis protein ASY1 and the meiotic recombinase DMC1, the euchromatic marks H3K4me1, H3K4me3 and H3K27ac, and the Polycomb mark H3K27me3, and non-significant negative correlations with H3K9me2, H3K27me1, centromeric CENH3, and DNA methylation in CG and CHG contexts. Therefore, while crossovers are increased in *fancm*, their genomic distribution is comparable to that in wild type. This suggests that chromatin may influence the recombination landscape in similar ways in both wild type and *fancm*. We have included a new heatmap showing these relationships in SFig. 10.

2. Probably I missed this point but there seem to be 5 intervals in which the recombination rate is negatively affected by the *fancm* mutations (Table 1), including one drastically affected on chr5B. Is there something special with these intervals in the wheat genome?

We did not find any specific feature in these five intervals which we could associate with the observed reduced recombination. These five regions are also not particularly remarkable in their gene content to allow speculation as to why recombination is reduced compared to the eleven intervals where recombination increases.

3. Lines 504:512: is the recombination rate in the tetraploid given as a control? The effect seems opposite to Table 1.

The tetraploid is not a control. We have analysed recombination in both the tetraploid and hexaploid with molecular markers to determine the effects of *fancm* on crossover formation at different ploidy levels. The rate of recombination in the tetraploid is different to the hexaploid through reasons that are not fully understood, possibly due to gene dosage. Based on our HEI10 data, the tetraploid loses more class I COs relative to the hexaploid and therefore the recombination rate is lower.

Minor point: tetraploid/hexaploid could be indicated on Table 1&2 legends, and the tables could be better detailed (mean etc.).

We have updated the table legends and clarified the average/means as requested.

Table 1. Recombination analysis in the hexaploid Avalon x Cadenza F₃ *fancm* mutant mapping population

Chr.	Markers	Marker Region	Position	Control	fancm	Delta	% Difference
1A_1	162	AX-94733072 - AX-94752690	2378195-332208434	93.76	126.60	32.84	35.0%
1A_2	12	AX-94874424 - AX-95151237	548494535-586914258	3.33	11.44	8.11	243.3%
1B_1	116	AX-95025932 - AX-94731046	571060391-645252540	67.91	83.51	15.61	23.0%
1B_2	13	AX-94813152 - AX-94804524	658531550-662842760	12.24	10.66	-1.58	-12.9%
2B	63	AX-95630341 - AX-94643790	2177758-52907517	46.89	40.27	-6.62	-14.1%
3B_1	19	AX-94653790 - AX-94633883	655306481-672099880	11.79	18.27	6.48	55.0%
3B_2	17	AX-95248765 - AX-94990660	821077029-829286625	7.34	11.99	4.65	63.3%
3D_1	15	AX-94534357 - AX-94667914	132604563-450294132	37.08	27.06	-10.02	-27.0%
3D_2	9	AX-95010529 - AX-94721213	564307274-571493665	19.87	20.92	1.05	5.3%
4A_1	10	AX-95159296 - AX-94999060	626119038-684968665	14.14	12.34	-1.80	-12.7%
4A_2	4	AX-94775503 - AX-94877844	713351841-723516355	10.52	20.58	10.06	95.6%
4A_3	32	AX-95153315 - AX-94555122	726214864-742002004	28.21	29.18	0.97	3.4%
5A_1	25	AX-94654340 - AX-95257149	607681540-619778700	5.34	9.57	4.24	79.4%
5A_2	24	AX-94995722 - AX-95230303	676592388-685543181	10.22	9.89	-0.33	-3.2%
5B_1	13	AX-95106649 - AX-95258242	535359886-544608954	20.53	4.51	-16.02	-78.0%
5B_2	74	AX-95151783 - AX-94657489	584127473-633943398	20.12	23.19	3.08	15.3%
5B_3	14	AX-94885467 - AX-94474369	693674319-708229373	18.74	20.17	1.43	7.6%
7A_1	48	AX-94485699 - AX-94664110	1704442-20294508	36.84	38.44	1.60	4.4%
7A_2	30	AX-94802515 - AX-94505773	61887949-83631459	22.92	26.30	3.38	14.8%
7B_1	22	AX-94733039 - AX-95191125	3338679-21668399	26.66	32.04	5.38	20.2%
7B_2	48	AX-94582074 - AX-95104628	724954096-750605855	29.68	70.99	41.31	139.2%
Average all intervals						4.94	31.3%
Total map length (cM)				544.13	647.92	103.79	19.1%

Table 2. Recombination analysis in the tetraploid Kronos F₃ *fancm* mutant mapping population

Chr.	Markers	Marker Region	Position	Control	fancm	Delta	% Difference
1B	2	K3842.1B.563292311- K3842.1B.591113626	563292311-591113626	10.3	13.6	3.3	32.00%
2A	2	K3842.2A.379735839- K3842.2A.651081672	379735839-651081672	21.8	16.3	-5.5	-25.40%
3A	2	K3842.3A.507462812- K3842.3A.692623524	507462812-692623524	27.7	30	2.3	8.20%
2B	2	K3873.2B.223733552- K3873.2B.486585051	223733552-486585051	10	6.5	-3.5	-34.80%
3B	3	K3873.3B.223037264- K3873.3B.723453707	223037264-723453707	24.7	26.6	1.9	7.80%
2B	2	K3842.2B.367667794- K3842.2B.545253995	367667794-545253995	9.9	8.8	-1	-10.60%
3A	2	K3873.3A.53209641- K3873.3A.423481141	53209641-423481141	13.5	11.5	-1.9	-14.30%
Average all intervals						-0.63	-5.30%
Total map length (cM)				117.9	113.3	-4.6	-3.9%

Concerning the genetic maps, I do not understand the last sentence L501-502. What is the difference between a cM map-length increase and a cumulative increase?

The '*cM map length increase*' of 19.1% is calculated as the percentage increase of the total *fancm* cM distance over WT control across all regions [from Table 1: $\Sigma_{\text{control}} = 544.1$ cM and $\Sigma_{\text{fancm}} = 647.91$ cM ; $(647.91-544.1)/ 544.1 = 19.1\%$], whereas the '*cumulative percentage increase*' describes the average of the percentage difference of all chromosome regions. We have also updated Table 1 and Table 2 for clarity.

We have revised the wording to clarify this.

The map length was increased by 19.1% from *fancm* over WT when adding the cM distances across all regions, or by 31.3% when averaging the percentage cM increase from the 21 intervals.

Reviewers' Comments:

Reviewer #2:

Remarks to the Author:

This is a revised version of the manuscript entitled "FANCM plays dual roles promoting class I interfering crossovers and suppressing class II non-interfering crossovers in wheat meiosis" by Desjardins et al.

Most of the concerns that were raised by all the referees have been taken into account, including mine. I still regret that the distribution of recombination in wheat is not more described in the introduction while there are numerous papers on the subject that have been published during the last 5-10 years. There is a simple sentence added citing a paper from the authors with a correlation with a histone mark which is marginal among all the literature while within this same paper, there are numerous analyses done.

Reviewer #4:

Remarks to the Author:

Overall, the authors have addressed my concerns. I'm not sure my first point was clear enough but this does not affect the quality of the manuscript.